# FINE-TUNING CLIP'S LAST VISUAL PROJECTOR: A FEW-SHOT CORNUCOPIA

## ABSTRACT

We consider the problem of adapting a contrastively pretrained vision-language model like CLIP (Radford et al., 2021) for few-shot classification. The existing literature addresses this problem by learning a linear classifier of the frozen visual features, optimizing word embeddings, or learning external feature adapters. This paper introduces an alternative way for CLIP adaptation without adding "external" parameters to optimize. We find that simply fine-tuning the last projection matrix of the vision encoder leads to strong performance compared to the existing baselines. Furthermore, we show that regularizing training with the distance between the fine-tuned and pretrained matrices adds reliability for adapting CLIP through this layer. Perhaps surprisingly, this approach, coined ProLIP, yields performances on par or better than state of the art on 11 few-shot classification benchmarks, few-shot domain generalization, cross-dataset transfer and test-time adaptation. Code will be made available online.

## 1 INTRODUCTION

Contrastive Language-Image Pretraining (CLIP) (Radford et al., 2021) has brought a breakthrough to visual representation learning, showing that strong visual features can be learned from noisy natural language descriptions at *very* large scale. The true potential of CLIP lies in its shared vision-text space, breaking the long-standing constraints of closed-set systems and enabling non-trivial interactions and querying between text and images via prompts. Such a freedom in the label space makes the model readily applicable to a wide range of specialized downstream applications.

CLIP trains both a vision and text encoders (we respectively denote $f$ and $g$) on large batches of image-text pairs using a sum of contrastive image-to-text and text-to-image losses. At inference, given an image $\mathsf{I}$, one only needs the names of $K$ candidate classes to perform *zero-shot classification*: $\hat{k} = \operatorname{argmax}_k \boldsymbol{v}^\mathsf{T}\boldsymbol{t}_k$ , where $\boldsymbol{v} = f(\mathsf{I}; \theta_f)/\|f(\mathsf{I}; \theta_f)\|_2$, $\boldsymbol{t}_k = g(\boldsymbol{T}_k; \theta_g)/\|g(\boldsymbol{T}_k; \theta_g)\|_2$; $\theta_f$ and $\theta_g$ are the frozen parameters of $f$ and $g$, respectively, $\boldsymbol{T}_k$ is a text prompt describing the class $k$, e.g., "a photo of $\{\text{class}_k\}$". The prompt template can be engineered to boost the zero-shot performance, or automated by querying multiple descriptors of a class from Large Language Model (LLMs), such as GPT-3 (Brown, 2020), and ensembling their embeddings (Menon & Vondrick, 2023). Yet, the zero-shot performance can still be unsatisfying, especially for data that are supposedly under-represented in CLIP training data. Examples of such cases include geospatial data, e.g., EuroSAT (Helber et al., 2019) and specialized data, e.g., FGVCAircraft (Maji et al., 2013). Thus, an interesting practical setting emerged in transfer learning: *Given a labeled few-shot training dataset of images, how to efficiently adapt CLIP in order to maximize the performance on the test set?*

Hinging on only a few labeled samples for supervision, model training is prone to overfitting. The common strategy is to avoid full fine-tuning and instead adapt only a few parameters (Kumar et al., 2022). Starting from a concept-rich pretrained CLIP model, such parameter-efficient strategies have been shown to be effective for few-shot tasks. In this direction, the literature explores three avenues. First, Context Optimization (CoOp) (Zhou et al., 2022b;a) parameterizes the template of $\boldsymbol{T}_k$ in the word embedding space, i.e., $\boldsymbol{T}_k = [\boldsymbol{w}]_1[\boldsymbol{w}]_2...[\boldsymbol{w}]_M[\text{class}_k]$ with $[\boldsymbol{w}]_1[\boldsymbol{w}]_2...[\boldsymbol{w}]_M$ learned while keeping $f$ and $g$ frozen. Second, CLIP adapters (Gao et al., 2024) learn a multi-layer perceptron (MLP) $h$ with a residual connection $\alpha$ on top of the frozen visual features $\boldsymbol{v}$, i.e., $\boldsymbol{v} := \alpha\boldsymbol{v} + (1 - \alpha)h(\boldsymbol{v})$. In both cases, the probability that a sample $i$ belongs to the class $k$ is $p_{ik} \propto \exp(\boldsymbol{v}_i^\mathsf{T}\boldsymbol{t}_k)$, meaning that the text embeddings are used as classification weights. Third, linear

probing (Radford et al., 2021) simply trains a linear classifier $\boldsymbol{W} \in \mathbb{R}^{D \times K}$ on top of the frozen visual features, $D$ being the embedding space dimension.

In all cases, the cross-entropy loss is used to train the set of parameters $\{\boldsymbol{w}\}$ using $N$ samples from each class $k$:

$$L(\{\boldsymbol{w}\}) = -\frac{1}{N} \sum_{i=1}^{N} \sum_{k=1}^{K} y_{ik} \log p_{ik}(\{\boldsymbol{w}\}). \tag{1}$$

**Shortcomings.** While existing solutions are technically simple and parameter-efficient, we identify several limitations. Prompt learning methods (Zhou et al., 2022b; Chen et al., 2023; Zhu et al., 2023) are slow to train as gradients need to be backpropagated over the entire text encoder. In addition, different context lengths and class-name positions lead to different performances. Adapters (Gao et al., 2024; Zhang et al., 2022) impose architectural choices of the MLP, the bottleneck dimension and the residual connection. On the other hand, while initially suggested as a few-shot baseline for CLIP, the performance of linear probing (LP) lies far behind adapters and prompt learning. Its main shortcoming stems from ignoring the text embeddings during adaptation. Recently, Huang et al. (2024) proposed LP++, an improved LP version blending textual embeddings with classification weights using class-wise learnable parameters. While LP++ (Huang et al., 2024) shows significant improvements over standard LP, we argue in this work that directly using the text embeddings as classification weights might be a better practice for fine-tuning CLIP, as it is closer in principle to its original pretraining regime.

The previous methods either train "external" parameters (e.g., Adapters, LP), or learn parameters in the input space (e.g., prompt learning). To the best of our knowledge, no existing work tackles few-shot CLIP adaptation problem with parameter-efficient fine-tuning of the model weights. In this work, we propose a first baseline for model weights based few-shot learning. Our method, dubbed "ProLIP", is both extremely simple to implement and effective: *considering pretrained CLIP encoders $f$ and $g$, we fine-tune the last visual projection matrix of $f$ (i.e., the projector mapping visual embeddings into the shared embedding space) with a cross-entropy loss (Equation 1) while constraining its weights to remain close to the pretrained ones.* More details are provided in Section 2.

ProLIP is advantageous for a number of reasons: **(1)** It alleviates the need of "external" parameters which usually imply architectural design search and/or heavy hyperparameter selection; **(2)** As backpropagation is only applied on the last projector of the vision encoder, training is fast, requiring only few seconds like LP++ (Huang et al., 2024); **(3)** ProLIP uses native text embeddings as classification weights in the few-shot task, which is aligned with the way CLIP is pretrained; **(4)** It balances pretraining and adaptation by imposing a simple regularizer based on l2-distance between weights. Our simple method performs better than the literature on few-shot adaptation and few-shot domain generalization, and is competitive on cross-dataset generalization and test-time adaptation.

## 2 PROLIP

### 2.1 PRELIMINARY ON CLIP ARCHITECTURE

CLIP adopts a transformer architecture for the text encoder, but the vision encoder may be either a ResNet (He et al., 2016) or a Vision Transformer (ViT) (Dosovitskiy et al., 2021). We detail both architectures below and later elaborate on our unified method applicable to both architectures regardless of their intrinsic differences.

**ResNet.** CLIP replaces the global average pooling layer in ResNet with an attention pooling layer. The output of the multi-head attention layer is then projected to the shared latent space using a linear layer. Thus, $f$ can be written as $f = f_1 \circ f_2$, where $f_1$ represents all the layers up to the attention pooling (included), and $f_2$ represents the linear projection head. Given an image $\mathsf{I}$:

$$\boldsymbol{x}_o = f_1(\mathsf{I}), \quad \boldsymbol{v} = f_2(\boldsymbol{x}_o) = \boldsymbol{W}_o^\mathsf{T} \boldsymbol{x}_o + \boldsymbol{b}_o, \tag{2}$$

$\boldsymbol{x}_o \in \mathbb{R}^{D_o}$ is the output of the attention pooling layer, $\boldsymbol{W}_o \in \mathbb{R}^{D_o \times D}$ is the projection matrix and $\boldsymbol{b}_o$ is a bias term.

**ViT.** The transformer encoder consists of multiple residual attention blocks. Each block has two main components: a multi-head self-attention and a feed-forward neural network (MLP), with residual connections. The output of the last residual attention block is projected to the latent space using

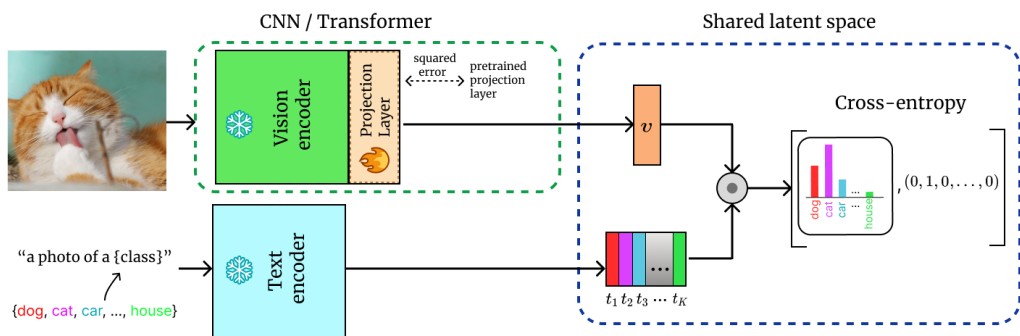

Figure 1: **ProLIP for few-shot adaptation.** Whether the vision encoder is a CNN or a transformer, ProLIP trains only the final linear layer that projects the visual embeddings into the shared latent space. The text encoder is frozen, and the text embeddings of the $K$ target concepts are used as classification weights. Training with cross-entropy is regularized with a squared error loss constraining the weights of the final projection to remain close to pretrained ones.

a trainable matrix. Thus, $f$ can be written as $f = f_1 \circ f_2$, where $f_1$ represents all the layers up to the last residual attention block (included), and $f_2$ represents the projection matrix. Given an image $\mathbf{I}$:

$$\boldsymbol{x}_o = f_1(\mathbf{I}), \quad \boldsymbol{v} = f_2(\boldsymbol{x}_o) = \boldsymbol{W}_o^\intercal \boldsymbol{x}_o, \tag{3}$$

where no bias term is included, unlike Equation 2.

## 2.2 THE LAST VISUAL PROJECTOR

We show that fine-tuning only the projection matrix $\boldsymbol{W}_o$ in Equations 2 or 3 can be a strong alternative to prompt learning and feature adapters. Specifically, the probability that a sample $i$ belongs to the class $k$ is computed as the Softmax over cosine similarities of image-text embeddings:

$$p_{ik}(\boldsymbol{W}_o) = \frac{\exp(\boldsymbol{v}_i^\intercal \boldsymbol{t}_k / \tau)}{\sum_{j=1}^{K} \exp(\boldsymbol{v}_i^\intercal \boldsymbol{t}_j / \tau)}, \tag{4}$$

$\boldsymbol{t}_k$ being fixed since $g$ is frozen, $\tau$ the pretraining temperature parameter, and $\boldsymbol{v}_i$ being a function of the input image, the frozen weights of $f$ and the learnable projection matrix $\boldsymbol{W}_o$. This matrix is learned with gradient descent using the cross-entropy loss $L(\boldsymbol{W}_o)$ defined in Equation 1.

**Regularization.** CLIP encoders map text and image modalities into a common latent space where strong image-text representation correspondences are established. We argue that unconstrained fine-tuning can lead to forgetting of the rich pretraining knowledge that appears through non-trivial zero-shot classification accuracies. Thus, a good fine-tuning strategy should balance pretraining knowledge preservation and adaptation to downstream task. Consequently, to prevent significant drift from the pretraining weights (i.e., knowledge forgetting), we regularize the training with the Frobenius norm of the difference between the pretrained and fine-tuned matrices. The total loss is:

$$\text{Loss} = L(\boldsymbol{W}_o) + \lambda \|\boldsymbol{W}_o - \boldsymbol{W}_o^{(0)}\|_{\text{F}}^2, \tag{5}$$

where $\boldsymbol{W}_o^{(0)}$ denotes the pretrained value of $\boldsymbol{W}_o$. We show later that $\lambda$ can be chosen as a decreasing function of the number of shots, as overfitting risk increases with less data (Hastie et al., 2009).

Algorithm 1 provides a PyTorch-like (Paszke et al., 2019) pseudo-code for ProLIP, representing one iteration of training. The method is illustrated in Figure 1.

## 3 EXPERIMENTS

**Datasets.** Following previous CLIP-based few-shot learning works, we experimentally test ProLIP on 11 classification datasets: ImageNet (Deng et al., 2009), SUN397 (Xiao et al., 2010),

DTD (Cimpoi et al., 2014), Caltech101 (Fei-Fei et al., 2004), UCF101 (Soomro, 2012), Flowers102 (Nilsback & Zisserman, 2008), StanfordCars (Krause et al., 2013), FGVCAircraft (Maji et al., 2013), EuroSAT (Helber et al., 2019), OxfordPets (Parkhi et al., 2012) and Food101 (Bossard et al., 2014).

For domain generalization experiments we follow ProGrad (Zhu et al., 2023), using ImageNet as source dataset and testing on ImageNet-V2 (Recht et al., 2019), ImageNet-Sketch (Wang et al., 2019), ImageNet-A (Hendrycks et al., 2021b) and ImageNet-R (Hendrycks et al., 2021a) as out-of-distribution datasets. For the cross-dataset transfer experiment, ProLIP is trained on ImageNet and evaluated on the other 10 datasets, similar to Prograd (Zhu et al., 2023).

**Training details.** We follow previous works and use $N \in \{1, 2, 4, 8, 16\}$ shots as support training set for few-shot classification. LP++ (Huang et al., 2024) identify a loophole in the few-shot CLIP literature (Zhang et al., 2022), which is the use of a large validation set for hyperparameter tuning. Instead, authors of LP++ propose using a validation set with as many shots as in the training set. For consistency, we adopt this protocol in our main experiments though also proposing a setting without any validation set. Huang et al. (2024) also remark that prior works evaluate their methods based on one or three support sets, leading to large standard deviations when the few-shot set is not representative of the class distribution. We follow their practice and evaluate ProLIP on 10 random seeds (i.e., support training sets) for each dataset. For domain generalization and cross-dataset transfer experiments, we select $N = 4$ like ProGrad (Zhu et al., 2023). Unless otherwise stated, we employ ResNet-50 with CLIP weights as the visual encoder, similarly to the literature. Training runs for 300 epochs on a full-batch of features, requiring only few seconds on a single Tesla V100 GPU. The learning rate (LR) and regularizer loss weight $\lambda$ are selected by grid search on the few-shot validation set, with LR in $\{10^{-2}, 10^{-3}, 10^{-4}, 10^{-5}, 10^{-6}, 10^{-7}, 10^{-8}\}$ and $\lambda$ in $\{10, 1, 10^{-1}, 10^{-2}, 10^{-3}, 10^{-4}, 0\}$.

We later show that using the regularizer ($\lambda > 0$) is better to avoid severe overfitting, and that state-of-the-art results can be still achieved with a fixed LR over all the datasets, and a fixed $\lambda$ which can be chosen as a decreasing function of the number $N$ of shots.

### 3.1 FEW-SHOT CLASSIFICATION

We compare ProLIP to baselines covering the variety of existing adaptation strategies. For prompt learning methods, we report CoOP (Zhou et al., 2022b) and its other variants PLOT (Chen et al., 2023), KgCoOp (Yao et al., 2023) and ProGrad (Zhu et al., 2023). For adapters, we compare to CLIP-adapter (Gao et al., 2024) and Tip-adapter (Zhang et al., 2022). Note that Tip-adapter performance is reported in two settings following (Huang et al., 2024): Tip-adapter-F where its two crucial hyperparameters are set to 1 and the validation set is used for early stopping, and Tip-adapter-F* where intensive hyperparameter search is performed to find the best values of the same hyperparameters based on the same validation set. For linear probing, we report LP (Radford et al., 2021) and LP++ (Huang et al., 2024). Baselines results are taken from (Huang et al., 2024) which employs early stopping for all the methods based on the validation set. For a fair comparison we follow the same experimental protocol (i.e., few-shot validation set, 10 random support sets) although we stress that we do not use early stopping but rather report the last model performance.

Table 1 reports the average classification accuracy and standard deviation, across 11 datasets. Per-dataset performances are reported in Appendix B. In all few-shots settings (i.e., $N \in \{1, 2, 4, 8, 16\}$), ProLIP clearly outperforms all the baselines, showing a great potential of the extremely simple approach of fine-tuning the last visual projector for adaptation. Moreover, we show in Section 3.2 that regularizing the projection matrix weights by minimizing their distance to the pretrained ones improves the classification accuracy. Besides, the additional merit of our regularization is to reduce sensitivity to hyperparameter selection; therefore paving the way for a more realistic few-shot adaptation setting detailed next.

For ProLIP, the statistics of the hyperparameters found by grid search (cf. Appendix C) show that performance-wise the best learning rates span a wide range of values. This is because our regularization term alleviates overfitting on the training set (cf. Section 3.2), therefore allowing the use of larger learning rate (e.g., $10^{-2}$) which would otherwise cause severe overfitting.

These results motivate the following question: *How sensitive is ProLIP performance to the choice of hyperparameters in a harder yet more realistic setting: Having no validation data at all?*

Table 1: **Few-shot image classification based on CLIP.** We report the classification accuracy (%) averaged over 11 datasets and 10 support sets, along with standard deviation, comparing ProLIP to other baseline adaptation methods. We highlight **best** and 2nd best. First row provides zero-shot classification accuracy for reference.

| Method | | $N = 1$ | 2 | 4 | 8 | 16 |
|---|---|---|---|---|---|---|
| Zero-shot CLIP | | | | 58.89 | | |
| Prompt Learning | CoOp | $59.62_{\pm3.11}$ | $63.80_{\pm2.32}$ | $67.23_{\pm1.64}$ | $71.30_{\pm0.86}$ | $74.06_{\pm0.55}$ |
| | PLOT | $61.51_{\pm2.91}$ | $65.67_{\pm2.06}$ | $68.39_{\pm1.17}$ | $71.96_{\pm0.70}$ | $74.35_{\pm0.66}$ |
| | KgCoOp | $61.36_{\pm3.04}$ | $63.23_{\pm2.06}$ | $65.73_{\pm1.15}$ | $67.50_{\pm1.11}$ | $69.01_{\pm0.79}$ |
| | ProGrad | $62.46_{\pm1.89}$ | $65.88_{\pm1.46}$ | $68.52_{\pm1.15}$ | $71.82_{\pm0.11}$ | $73.95_{\pm0.68}$ |
| Adapters | CLIP-Adapter | $60.32_{\pm0.80}$ | $61.93_{\pm0.93}$ | $65.12_{\pm0.80}$ | $69.20_{\pm0.56}$ | $72.57_{\pm0.54}$ |
| | Tip-Adapter-F | $61.29_{\pm0.92}$ | $62.94_{\pm0.75}$ | $66.02_{\pm0.80}$ | $69.88_{\pm0.51}$ | $73.82_{\pm0.55}$ |
| | Tip-Adapter-F* | $63.06_{\pm1.05}$ | $66.47_{\pm0.65}$ | $68.71_{\pm0.96}$ | $71.78_{\pm1.00}$ | $74.37_{\pm0.35}$ |
| Linear Probing | LP | $36.10_{\pm1.43}$ | $46.99_{\pm1.29}$ | $56.72_{\pm1.20}$ | $64.66_{\pm0.55}$ | $70.56_{\pm0.44}$ |
| | LP++ | $63.43_{\pm0.90}$ | $66.20_{\pm0.72}$ | $69.16_{\pm0.79}$ | $72.04_{\pm0.46}$ | $74.42_{\pm0.45}$ |
| Model weights | ProLIP | **$64.59_{\pm0.98}$** | **$67.09_{\pm0.87}$** | **$70.53_{\pm0.69}$** | **$73.40_{\pm0.45}$** | **$76.55_{\pm0.41}$** |

Figure 2: **ProLIP sensitivity to hyperparameter choice.** Accuracy of ProLIP to the hyperparameters (learning rate and regularization weight $\lambda$) for $N \in \{1, 2, 4, 8, 16\}$-shot settings. Each data point is an average over 11 datasets, 10 runs for each.

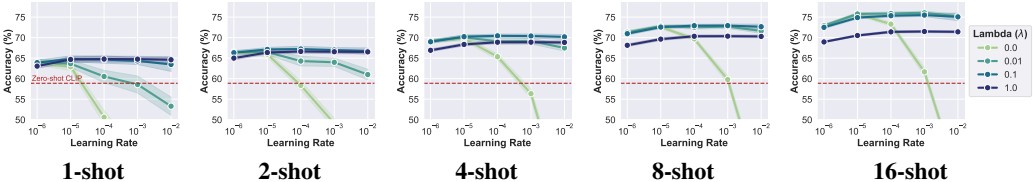

**1-shot**        **2-shot**        **4-shot**        **8-shot**        **16-shot**

## 3.2 TOWARDS A MORE REALISTIC FEW-SHOT SETTING WITH NO VALIDATION SET

In contrast to prior practices, we also propose a setting which does not rely on any validation set. The main motivation for few-shot classification being propelled by the labeled data scarcity in real-world situations, we argue that relying even on $N$-shot validation sets as in (Huang et al., 2024) may be seen as a violation of the $N$-shot setting since it effectively requires access to $2N$ examples ($N$ for training and $N$ for validation). It follows that the true merit of a method stems from its lower sensitivity to hyperparameter choice.

The importance of our proposed regularization loss appears by testing ProLIP with different fixed hyperparameters. Figure 2 shows the average accuracy achieved by ProLIP across the 11 datasets, for 5 different LR values combined with regularization ($\lambda \in \{10^{-2}, 10^{-1}, 1\}$) or without regularization ($\lambda = 0$). For $\lambda = 0$, the accuracy drops dramatically when the learning rate is greater than $10^{-5}$. On the other hand, using the weight regularizer ($\lambda > 0$), ProLIP training becomes intriguingly less prone to overfitting. Even for high learning rate, different fixed values of $\lambda$ still prevent the accuracy degradation observed when no regularization is applied. Thus, we argue that our regularized method is a principled approach towards a trustworthy training of CLIP on few-shot setting in real-world scenarios. We refer to Table 8 in appendices for detailed performances. We observe that for $N = 1$ and $N = 2$, fixed values of the hyperparameters across all the 11 datasets yield better average accuracy than performing grid search. This observation is not surprising, especially for these two extreme low-shot settings, as the performance on a validation set having the same degree of scarcity can be noisy and not representative of the larger test set.

$\lambda$ **as function of $N$.** It can be observed from Figure 2 and Table 8 that for lower-shot settings, higher $\lambda$ values lead to better accuracy, and vice versa. Thus, one can benefit from this expected observation by formulating $\lambda$ as a decreasing function of the number of shots $N$, constituting a step towards principled and realistic few-shot classification evaluation. Table 2 corroborates our proposition: for different values of learning rate, $\lambda$ expressed as a simple function of $N$ (e.g., $1/N$,

Table 2: **ProLIP with a parametric $\lambda$.** Accuracy (%) of ProLIP with fixed learning rate (LR) and $\lambda$ as a function of $N$. For each $\lambda$ value, we report performance for different LRs and averaged across LRs. Numbers are averages over 11 datasets and 10 runs. We highlight **best** and 2nd best for averages across LRs.

| Method | | $N = 1$ | 2 | 4 | 8 | 16 |
|---|---|---|---|---|---|---|
| Zero-shot CLIP | | | | 58.89 | | |
| ProLIP, $\lambda = 1/N$ | LR=$10^{-5}$ | $64.67_{\pm0.63}$ | $66.80_{\pm0.39}$ | $69.73_{\pm0.37}$ | $72.44_{\pm0.33}$ | $75.34_{\pm0.31}$ |
| | LR=$10^{-4}$ | $64.78_{\pm0.67}$ | $67.02_{\pm0.46}$ | $70.19_{\pm0.36}$ | $72.83_{\pm0.35}$ | $75.65_{\pm0.35}$ |
| | LR=$10^{-3}$ | $64.75_{\pm0.68}$ | $66.93_{\pm0.49}$ | $70.12_{\pm0.39}$ | $72.84_{\pm0.34}$ | $75.80_{\pm0.35}$ |
| | LR=$10^{-2}$ | $64.60_{\pm0.72}$ | $66.93_{\pm0.56}$ | $70.03_{\pm0.50}$ | $72.63_{\pm0.48}$ | $75.08_{\pm2.06}$ |
| | Average | $\mathbf{64.70}_{\pm0.08}$ | $\underline{66.92}_{\pm0.09}$ | $\underline{70.02}_{\pm0.20}$ | $\mathbf{72.69}_{\pm0.19}$ | $\mathbf{75.47}_{\pm0.32}$ |
| ProLIP, $\lambda = 1/N^2$ | LR=$10^{-5}$ | $64.67_{\pm0.63}$ | $67.04_{\pm0.42}$ | $70.27_{\pm0.44}$ | $72.84_{\pm0.32}$ | $75.77_{\pm0.30}$ |
| | LR=$10^{-4}$ | $64.78_{\pm0.67}$ | $67.22_{\pm0.50}$ | $70.42_{\pm0.43}$ | $72.78_{\pm0.38}$ | $75.16_{\pm0.37}$ |
| | LR=$10^{-3}$ | $64.75_{\pm0.68}$ | $67.12_{\pm0.52}$ | $70.42_{\pm0.52}$ | $72.89_{\pm0.40}$ | $75.85_{\pm0.41}$ |
| | LR=$10^{-2}$ | $64.60_{\pm0.72}$ | $67.05_{\pm0.55}$ | $70.09_{\pm0.81}$ | $72.19_{\pm0.42}$ | $74.54_{\pm0.71}$ |
| | Average | $\mathbf{64.70}_{\pm0.08}$ | $\mathbf{67.11}_{\pm0.08}$ | $\mathbf{70.30}_{\pm0.16}$ | $\underline{72.68}_{\pm0.33}$ | $\underline{75.33}_{\pm0.61}$ |
| ProLIP, $\lambda = 0$ | LR=$10^{-5}$ | $62.91_{\pm0.85}$ | $65.97_{\pm0.67}$ | $69.76_{\pm0.55}$ | $72.57_{\pm0.35}$ | $75.73_{\pm0.30}$ |
| | LR=$10^{-4}$ | $50.61_{\pm1.60}$ | $58.36_{\pm1.05}$ | $65.35_{\pm0.67}$ | $69.70_{\pm0.45}$ | $73.29_{\pm0.47}$ |
| | LR=$10^{-3}$ | $40.17_{\pm1.65}$ | $49.08_{\pm1.19}$ | $56.34_{\pm1.08}$ | $59.78_{\pm0.92}$ | $61.67_{\pm0.96}$ |
| | LR=$10^{-2}$ | $20.02_{\pm2.21}$ | $23.91_{\pm2.17}$ | $28.05_{\pm2.43}$ | $32.02_{\pm1.93}$ | $35.75_{\pm1.25}$ |
| | Average | $43.43_{\pm18.16}$ | $49.33_{\pm18.30}$ | $54.88_{\pm18.74}$ | $58.52_{\pm18.50}$ | $61.61_{\pm18.30}$ |

$1/N^2$) leads to strong and stable results, with averages outperforming the state of the art. When no regularization is used ($\lambda = 0$), ProLIP is extremely sensitive to the learning rate choice and shows very high variances. *Note that we do not claim to solve few-shot setting without validation, but rather aim to demonstrate that ProLIP, with its regularization loss, is a strong candidate for such scenario due to its reduced sensitivity to hyperparameter choices.*

## 3.3 GENERALIZATION IN FEW-SHOT SETTINGS

**Domain generalization.** Real-world scenarios impose an additional challenge of distribution shift for model adaptation. Supposing the test data to follow the same distribution as the training is often unrealistic, and a model can be of practical interest only if it exhibits resilient generalization capabilities when confronted with out-of-distribution data. Achieving this generalization in a few-shot framework is highly challenging yet important to benchmark when assessing the potential practical use of CLIP. Following ProGrad, we train ProLIP on ImageNet (IN) as source dataset (with the number of shots $N = 4$), and assess it on ImageNet-V2 (IN-V2), ImageNet-Sketch (IN-S), ImageNet-A (IN-A) and ImageNet-R (IN-R). Table 3 shows that ProLIP is on par or better than other methods both on source and unseen target domains, for both ResNet and ViT CLIP backbones.

**Cross-dataset generalization.** An interesting question was introduced in prompt learning methods (Zhou et al., 2022a; Zhu et al., 2023): *How does a prompt learned from a single few-shot dataset perform when tested on other datasets?* This setting is referred to as cross-dataset transfer / generalization. Here, we ask the same question, i.e., whether the fine-tuned weights of the projection matrix can generalize across datasets. Table 4 shows the generalization from ImageNet as source dataset (4-shot) to the 10 other datasets. ProLIP outperforms ProGrad on 6 out of 11 datasets and on average. However, it is worth noting that zero-shot CLIP remains the strongest baseline in this setting. As argued in CoCoOp (Zhou et al., 2022a), ImageNet contains 1000 classes, mainly consisting of objects. Dog breeds are also present, so generalization (or at least small zero-shot performance drop) to datasets like OxfordPets and Caltech101 is expected. However, for datasets presenting a larger gap (e.g., fine-grained and/or specialized datasets), generalization is less expected. For such datasets, like FGVCAircraft and DTD, ProLIP outperforms other adaptation methods, but remains behind zero-shot accuracy. Among all the target datasets, all methods exhibit a significant drop on EuroSAT (26-35% from the zero-shot model). Interestingly, ProLIP not only does not exhibit the same drop, but retains the zero-shot performance. In short, looking at the generalization of ProLIP on each of the 10 datasets, our method is overall retaining zero-shot capability the most and showing cross-dataset transferability.

Table 3: **Domain generalization.** 4-shot training on ImageNet (source) and evaluation on out-of-distribution (OOD) variants with different visual backbones. Baselines are average of 3 runs reported from Prograd (Zhu et al., 2023).

| Backbone | Method | IN | IN-V2 | IN-S | IN-A | IN-R | Average | Avg. OOD |
|---|---|---|---|---|---|---|---|---|
| | Zero-shot CLIP | 66.73 | 60.84 | 46.13 | 47.80 | 74.01 | 59.10 | 57.20 |
| ViT-B/16 | LP | 54.70 | 45.57 | 28.20 | 22.47 | 44.12 | 39.01 | 35.09 |
| | CoOp | 69.86 | 62.83 | 46.90 | 48.98 | 74.55 | 60.62 | 58.32 |
| | CoCoOp | 70.13 | 63.05 | 46.48 | 49.36 | 73.80 | 60.56 | 58.17 |
| | Prograd | **70.45** | 63.35 | 48.17 | 49.45 | 75.21 | 61.33 | 59.05 |
| | ProLIP | 70.40 | **63.63** | **48.84** | **50.94** | **77.99** | **62.36** | **60.35** |
| | Zero-shot CLIP | 62.00 | 54.75 | 40.82 | 29.59 | 66.01 | 50.63 | 47.79 |
| ViT-B/32 | LP | 46.77 | 39.12 | 20.32 | 16.32 | 39.48 | 32.40 | 28.81 |
| | CoOp | 64.74 | 56.59 | 40.03 | 31.10 | 64.50 | 51.39 | 48.06 |
| | CoCoOp | 64.63 | 56.59 | 40.74 | 30.27 | 64.12 | 51.27 | 47.93 |
| | Prograd | 65.36 | 57.42 | 41.73 | **31.89** | 66.53 | 52.59 | 49.39 |
| | ProLIP | **65.49** | **57.52** | **42.40** | 31.91 | **69.82** | **53.43** | **50.41** |
| | Zero-shot CLIP | 58.18 | 51.34 | 33.32 | 21.65 | 56.00 | 44.10 | 40.58 |
| RN50 | LP | 41.29 | 33.65 | 13.09 | 11.18 | 26.82 | 25.21 | 21.19 |
| | CoOp | 61.34 | 53.81 | 32.83 | 22.08 | 54.62 | 44.94 | 40.84 |
| | CoCoOp | 61.04 | 53.71 | 32.30 | 22.07 | 53.60 | 44.54 | 40.42 |
| | Prograd | 62.17 | 54.70 | 34.40 | **23.05** | 56.77 | 46.22 | 42.23 |
| | ProLIP | **62.37** | **54.84** | **34.83** | 23.04 | **60.27** | **47.07** | **43.25** |
| | Zero-shot CLIP | 61.24 | 54.82 | 38.66 | 28.03 | 64.34 | 49.42 | 46.46 |
| RN101 | LP | 47.01 | 38.46 | 19.09 | 16.33 | 39.43 | 32.06 | 28.33 |
| | CoOp | 63.99 | 56.99 | 39.40 | 29.50 | 64.04 | 50.78 | 47.48 |
| | CoCoOp | 63.59 | 56.98 | 39.16 | 29.09 | 64.14 | 50.59 | 47.34 |
| | Prograd | 64.98 | **57.86** | 40.53 | **30.13** | 65.61 | 51.82 | 48.53 |
| | ProLIP | **65.13** | 57.52 | 40.66 | 30.12 | **67.05** | **52.10** | **48.84** |

Table 4: **Cross-dataset generalization.** Training is performed on 4-shot ImageNet (source). The learned models are evaluated on 10 other datasets (target). Baselines are average of 3 runs reported from Prograd (Zhu et al., 2023).

| Method | Source | Target | | | | | | | | | | |
| | ImageNet | Caltech101 | OxfordPets | StanfordCars | Flowers102 | Food101 | FGVCAircraft | SUN397 | DTD | Eurosat | UCF101 | Average |
|---|---|---|---|---|---|---|---|---|---|---|---|---|
| Zero-shot | 60.35 | 85.84 | 85.75 | **55.78** | 65.98 | **77.35** | 17.07 | 58.85 | **42.69** | 36.22 | 61.80 | **58.88** |
| CoOp | 61.34 | 84.48 | 85.99 | 54.16 | 60.10 | 75.48 | 14.09 | 57.48 | 35.32 | 26.72 | 57.56 | 55.70 |
| CoCoOp | 61.04 | 84.73 | 86.42 | 52.34 | 61.24 | 73.79 | 13.74 | 55.94 | 36.60 | 23.46 | 57.97 | 55.21 |
| Prograd | 62.17 | **88.30** | 86.43 | 55.61 | 62.69 | 76.76 | 15.76 | **60.16** | 39.48 | 24.87 | 58.70 | 57.36 |
| ProLIP | **62.37** | 87.08 | 84.57 | 54.56 | 64.79 | 75.56 | 16.94 | 59.78 | 40.96 | **36.31** | 61.24 | 58.56 |

## 3.4 BEYOND SUPERVISED ADAPTATION: TEST-TIME PROLIP

In this section, our goal is to show that ProLIP can be applied beyond supervised few-shot CLIP adaptation. Motivated by the risk of "overfitting" the source domain in classic prompt tuning methods (Zhou et al., 2022b;a), Shu et al. (2022) pioneered test-time prompt tuning (TPT), aiming to learn adaptive prompts on the fly using a single test image.

**TPT background knowledge.** TPT aims to learn a context specific to each test image in an unsupervised way. Given an unlabeled test image $I_{\text{test}}$, the prompt is learned by minimizing the average prediction entropy over different augmented views of $I_{\text{test}}$. Moreover, *confidence selection* filters out the augmented views with high entropy predictions, which might lack important information for classification. More details are provided in Appendix E.

**Test-time ProLIP.** We do not introduce a new way for CLIP test-time adaptation but simply follow the same experimental setting as TPT (i.e., 1-step entropy minimization of averaged prediction

probability distribution, confidence selection), although ProLIP optimizes the projection weight matrix $W_o$ instead of the prompt as in TPT. Table 5 shows that ProLIP yields competitive results to TPT on ImageNet and natural distribution shifts, while being one order of magnitude faster to train. Note that ProLIP uses the template "a photo of a {class}" for the text prompts. Instead, few-shot CLIP works (Zhang et al., 2022; Huang et al., 2024) use the average of 7 templates per class for ImageNet ("itap of a {class}.", "a bad photo of the {class}.", "a origami {class}.", "a photo of the large {class}.", "a {class} in a video game.", "art of the {class}.", "a photo of the small {class}."). As shown in Table 5, using these templates already boosts zero-shot classification accuracy. With almost no extra effort, using templates complementarily improves the performance of ProLIP, significantly outperforming TPT. For direct comparison, we separate the CoOp and TPT+CoOp results as those require tuning on ImageNet using 16-shot training data. Of note, our method still outperforms TPT+CoOp, while it does not use any labeled training data.

Table 5: **Robustness to natural distribution shifts in test-time adaptation.** Zero-shot CLIP, TPT and ProLIP do not require training data. CoOp and TPT+CoOp require 16-shot ImageNet training. Experiments are done with RN50 backbone.

| Method | IN | IN-A | IN-V2 | IN-R | IN-S | Average | Avg. OOD |
|---|---|---|---|---|---|---|---|
| Zero-shot CLIP | 58.16 | 21.83 | 51.41 | 56.15 | 33.37 | 44.18 | 40.69 |
| Zero-shot CLIP w/ templates | 60.33 | 23.79 | 53.31 | 60.58 | 35.46 | 46.69 | 43.29 |
| TPT | 60.74 | 26.67 | 54.70 | 59.11 | 35.09 | 47.26 | 43.89 |
| ProLIP | 60.00 | 30.57 | 54.09 | 58.29 | 35.13 | 47.62 | 44.52 |
| ProLIP w/ templates | **62.00** | **33.76** | **56.03** | **62.69** | **37.29** | **50.35** | **47.44** |
| CoOp | 63.33 | 23.06 | 55.40 | 56.60 | 34.67 | 46.61 | 42.43 |
| TPT + CoOp | 64.73 | 30.32 | 57.83 | 58.99 | 35.86 | 49.55 | 45.75 |

## 4 RELATED WORK

**Parameter-efficient fine-tuning (PEFT).** The advent of increasingly larger pretrained vision foundation models with excellent generalization capabilities has opened the way to new transfer learning approaches towards downstream tasks with limited labeled data. Full fine-tuning of such models turns out to be not only computationally inefficient but also often underperforming, even when compared to linear probing (Kumar et al., 2022; Wei et al., 2024). Parameter-efficient fine-tuning methods aim to adapt models effectively with minimal changes of their parameters while freezing most of the large pretrained backbone. Side-tuning (Zhang et al., 2020) trains a small network in parallel to a frozen pretrained network and avoids catastrophic forgetting. Optimizing only a specific subset of parameters of a model, e.g., bias terms (Zaken et al., 2022), is also an effective strategy. However this still requires full backpropagation through the pretrained model. Adapter-tuning methods add lightweight modules to transformer layers (Houlsby et al., 2019; Rücklé et al., 2021), but incur a higher runtime cost. LoRA (Hu et al., 2022) optimizes new low-rank matrices injected to transformer layers to approximate weight changes during fine-tuning, reducing significantly the number of parameters to learn. Prompt-tuning approaches, such as VPT (Jia et al., 2022), add a set of learnable prompts to the set of input patch embeddings. In addition to the additional computational cost for the full backpropagation and for runtime for most of them, these methods are specifically devised for transformer layers and are not directly applicable to convolutional networks.

**Few-shot CLIP adaptation**. CLIP's specific interaction between text and image features has enabled new adaptation methods fully exploiting this property in particular for the few-shot regime. Inspired by prompt learning in natural language processing (Zhong et al., 2021; Li & Liang, 2021), Zhou et al. (2022b) proposed context optimization (CoOp) which applies the same concept for pretrained vision-language models. CoOp was later shown not to generalize well on unseen classes within the same dataset. Thus, conditional context Optimization (CoCoOp) (Zhou et al., 2022a) adds a meta-network that generates input-conditional tokens in addition to the learnable vectors, making optimized context less prone to overfitting to the seen classes. Zhu et al. (2023) identified a critical problem of unconstrained prompt learning methods: in extreme low-shot settings, overfitting can even decrease the zero-shot performance. Consequently, they proposed regularizing the training by only updating prompts whose gradients do not conflict the direction resulting from zero-shot predictions. PLOT (Chen et al., 2023) applies optimal transport on sets of text and visual features to

learn the transport plan between the two sets in an inner loop, which is fixed in the outer loop where prompts are learned.

Instead of adapting the model in the input space, CLIP-adapter (Gao et al., 2024) adds an MLP on top of the features in the shared embedding space, with a residual connection to preserve the pretrained knowledge. Zhang et al. (2022) showed that training-free adapters can be competitive to trained ones. They create a cache-model for the few-shot training set from the visual features and the corresponding ground-truth labels, which are converted to the weights of the MLP adapter. Following the success of training-free CLIP adaptation, Wang et al. (2024) resorted to Gaussian Discriminative Analysis (GDA) (Bishop & Nasrabadi, 2006) that assumes the conditional distribution of features given the labels follows a multivariate normal distribution. Thus, they construct the GDA classifier using the mean vectors of each class and the inverse covariance matrix, and show that classification can be further improved by ensembling GDA and zero-shot classifiers. Our ProLIP takes advantage of CLIP's compatibility between text and images and adapts it to downstream tasks without additional learnable parameters or architecture changes.

## 5 CONCLUSION

In this work, we proposed an extremely simple way for efficient adaptation of CLIP based on the model weights. We identify that fine-tuning the last visual projection matrix, which projects the visual embedding into the multi-modal latent space, is a strong alternative for few-shot classification with CLIP. Moreover, we showed advantages of including a squared error regularizer preventing the drift from pretrained weights in making our method less sensitive to hyperparameters choice, making it a trustworthy candidate for realistic few-shot adaptation. Additionally, we provided experimental corroboration on the competitiveness of ProLIP in few-shot classification, domain generalization, cross-dataset transfer, and even test-time adaptation, making it a potential general framework for further applications.

**Limitations.** Similarly to prompt learning methods and unlike adapters and linear probing, ProLIP does not operate in a black-box setting. Many recent pretrained models are only available through APIs (e.g., GPT, Claude, Gemini), where practitioners can only get access to the end-point of the encoders (i.e., the latent embeddings). ProLIP does not apply to such closed models.

**Future directions.** Our framework can be applied to other foundation models with different modalities, downstream tasks and training objectives (Li et al., 2022; Zhai et al., 2023). Future research can explore other alternatives for model weights based adaptation, and theoretical investigation on the effect of fine-tuning the last visual projector.

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

APPENDIX

## A   ALGORITHM

Algorithm 1 provides a PyTorch-like pseudo-code for ProLIP, representing one iteration of training. The input features $x_o$ are computed by forwarding the augmented images through the frozen visual encoder up to the last projection matrix $W_o$ (excluded), then saved. Note that we use the same data augmentation as previous baselines (Zhang et al., 2022; Huang et al., 2024): `RandomResizedCrop` with a scale between 0.5 and 1 and `RandomHorizontalFlip` with a probability of 0.5.

On the text side, the text encoder is fully frozen. The templates are similar to previous works (Zhang et al., 2022; Huang et al., 2024) for a fair comparison, and are detailed below for each dataset.

Caltech101, StanfordCars, SUN397: `[a photo of a {class}.]`
DTD: `[{class} texture.]`
Eurosat: `[a centered satellite photo of {class}.]`
FGVCAircraft: `[a photo of a {class}, a type of aircraft.]`
Food101: `[a photo of {class}, a type of food.]`
Flowers102: `[a photo of a {class}, a type of flower.]`
OxfordPets: `[a photo of a {class}, a type of pet.]`
UCF101: `[a photo of a person doing {class}.]`

For ImageNet, also following previous works, we adopt an ensemble of 7 templates: `[itap of a {class}.]`, `[a bad photo of the {class}.]`, `[a origami {}.]`, `[a photo of the large {class}.]`, `[a {class} in a video game.]`, `[art of the {class}.]`, `[a photo of the small {class}.]`

For training, only the weight matrix $W_o$ is fine-tuned. Note that for ResNets, a bias term $b_o$ exists while for ViTs no bias is added in pretraining. We stress that fine-tuning also the bias term for ResNets does not change the results, as most of the parameters as concentrated in the weight matrix. In details for ResNet-50, $W_o \in \mathbb{R}^{D_o \times D}$, where $D_o = 2048$ and $D = 1024$, making a total of $\sim$2M parameters, while $b_o \in \mathbb{R}^D$ has only 1024 parameters.

## B   PER-DATASET FEW-SHOT CLASSIFICATION PERFORMANCE

To complement the average across datasets reported in Table 1, we also detail the *per-dataset performance* of all methods in Tables 6 and 7, reporting for each the average accuracy over 10 seeds (i.e., support sets). ProLIP performs particularily well on DTD, UCF101, StanfordCars, FGVCAircraft and EuroSAT. For some specific settings, e.g., 1-shot DTD, 1-shot EuroSAT, 16-shot StanfordCars, 8 and 16-shot FGVCAircraft, the improvements over state-of-the-art are significant. On the other hand, for datasets like OxfordPets and Food101, where the zero-shot performance is already good, ProLIP and other baselines are outperformed by prompt learning methods (e.g., ProGrad), which can be related to the relatively lower number of parameters in the latter, making them less prone to overfitting.

Future research may include the zero-shot accuracy on the few-shot training set in the parametric formulation of the regularization loss weight (i.e., $\lambda$). That is, the higher the zero-shot accuracy, the less should be the distance between the fine-tuned projection matrix and the pretrained one (i.e., higher $\lambda$).

## C   GRID SEARCH HYPERPARAMETERS

Figure 3 shows the distribution of hyperparameters found by grid search on the few-shot validation set (cf. Table 1). We draw two observations:

1. The learning rates span a wide range of values, and high values like $10^{-3}$ and $10^{-2}$ are selected several times, which would cause severe overfitting when no regularization is used (cf. Table 2 and Figure 2).

---

**Algorithm 1** PyTorch-like pseudo-code for ProLIP.

---

```python
# N : Number of shots
# model_init : CLIP pretrained model
# state_dict : weights of the pretrained model
# xo: input to the last projection matrix (N*K, Do)
# text_weights: normalized embeddings of classnames (K,D)

mse = nn.MSELoss(reduction='sum')

for name, param in model_init.named_parameters():
    param.requires_grad = False # Freeze all CLIP model parameters

if backbone == "ResNet":
    Wo = nn.Parameter(model_init.visual.attnpool.c_proj.weight)
    bo = nn.Parameter(model_init.visual.attnpool.c_proj.bias)

elif backbone == "ViT":
    Wo = nn.Parameter(state_dict["visual.proj"])
    bo = nn.Parameter(torch.zeros(D))

Wo_copy = copy.deepcopy(Wo) # Copy initial weights for use in the regularization loss
Wo.requires_grad = True     # Compute the gradient over the last projection matrix
bo.requires_grad = False

v = xo @ Wo + bo
v = F.normalize(v,dim=-1) #normalize the embeddings

logits = 100. * v @ text_weights.T #compute the cosine similarity scores

initial_params = Wo_copy.view(-1)
fine_tuned_params = Wo.view(-1)

loss = F.cross_entropy(logits, target) + lmda * mse(initial_params, fine_tuned_params)

optimizer.zero_grad()
loss.backward()
optimizer.step()
scheduler.step()
```

---



Figure 3: **Hyperparameters selected by grid search.** Learning rates and regularization loss weights $\lambda$ found with grid search on the few-shot validation set. The distribution of these hyperparameters are shown for each few-shot setting ($N = 1, 2, 4, 8, 16$).

2. $\lambda = 0$ is rarely selected, meaning that based on the few-shot validation set, regularized projection matrices generalize better.

## D PROLIP SENSITIVITY TO HYPERPARAMETERS

Table 8 complements Figure 2, where ProLIP is trained for different fixed learning rates, with fixed regularization loss weight values (i.e., $\lambda$). Looking at the values, we make the following observations:

1. For low learning rates (i.e., $10^{-5}$, $10^{-6}$), unregularized ProLIP shows good performance for different values of $N$, demonstrating the effectiveness of simply fine-tuning the last visual projection matrix. However, the performance drops significantly when the LR is lowered.

2. A higher value of $\lambda$ works better for fewer training shots $N$, and vice versa. This effect is increasingly visible when the LR increases. Such observation is expected: with less data we need more regularization as overfitting risk is higher, and is the base for formulating $\lambda$ as a decreasing function of $N$ (See Table 2).

## E  DETAILS ON TEST-TIME PROLIP

TPT (Shu et al., 2022) learns a single prompt for each test image using an unsupervised loss function. Given a test image $\mathbf{I}_{\text{test}}$, the image is augmented $N_{\text{views}}$ times using a family of random augmentations $\mathcal{A}$. Predictions are made for each view, and the training consists of minimizing the entropy of the averaged probability distribution of these predictions:

$$\boldsymbol{p}^* = \text{argmin}_{\boldsymbol{p}} - \sum_{i=1}^{K} \tilde{p}_{\boldsymbol{p}}(y_i|\mathbf{I}_{\text{test}}) \log \tilde{p}_{\boldsymbol{p}}(y_i|\mathbf{I}_{\text{test}}), \tag{6}$$

where

$$\tilde{p}_{\boldsymbol{p}}(y_i|\mathbf{I}_{\text{test}}) = \frac{1}{N_{\text{views}}} \sum_{i=1}^{N_{\text{views}}} p_{\boldsymbol{p}}(y_i|\mathcal{A}_i(\mathbf{I}_{\text{test}})). \tag{7}$$

In addition, *confidence selection* is used to filter out predictions with high entropy, which are considered as noisy. Self-entropy is computed for each of the $N_{\text{views}}$, a fixed cutoff percentile $\rho$ keeps only predictions with lower entropy than $\tau$. $\tilde{p}_{\boldsymbol{p}}$ in Equation 6 becomes:

$$\tilde{p}_{\boldsymbol{p}}(y|\mathbf{I}_{\text{test}}) = \frac{1}{\rho N} \sum_{i=1}^{N_{\text{views}}} \mathbb{1}_{\{H(p_i) \leq \tau\}} p_{\boldsymbol{p}}(y|\mathcal{A}_i(\mathbf{I}_{\text{test}})) \tag{8}$$

We apply the same framework (i.e., loss function, confidence selection) with the only difference of minimizing Equation 6 over $\boldsymbol{W}_o$ instead of the prompt $\boldsymbol{p}$. For a fair comparison, we use the same number of steps for training (i.e., 1 step) and the same value of the cutoff percentile $\rho = 0.1$. Note that, measured on ImageNet, ProLIP is $\sim 13$ times faster than TPT, as the latter requires backpropagation trough the whole text encoder, while in our case backpropagation is limied to the last visual projection layer and is not applied on the text encoder. We also stress that since we perform only 1 step of training, the regularization loss cannot be used as the first value it takes is $0$ (initially the fine-tuned projection matrix is equal to the pre-trained one).

## F  VISUALIZATION

We use UMAP to visualize EuroSAT test set feature manifolds, before and after 16-shot training (i.e., zero-shot *vs* ProLIP). The results are illustrated in Figure 4. We observe that the features are generally better clustered for ProLIP. Confusing categories like *Highway or Road, Permanent Crop Land and Pasture Land* exhibit remarkably better separation for our few-shot adapted model compared to zero-shot. This visualization hints that ProLIP learns better feature manifolds in the few-shot classification setting.

## G  COMPARISON WITH FULL FINE-TUNING

Here we compare ProLIP with full fine-tuning of the visual backbone. Results in Table 9 show that full fine-tuning is far behind ProLIP, and even degrades zero-shot performance for $N = 1, 2$ and 4-shots. The learning rate is $10^{-5}$ for these experiments, and ProLIP is shown for different $\lambda$ values (including $\lambda = 0$). These results confirm that full fine-tuning faces a high risk of overfitting especially in low-shot regimes, advocating for the importance of parameter-efficient fine-tuning methods, like ProLIP.

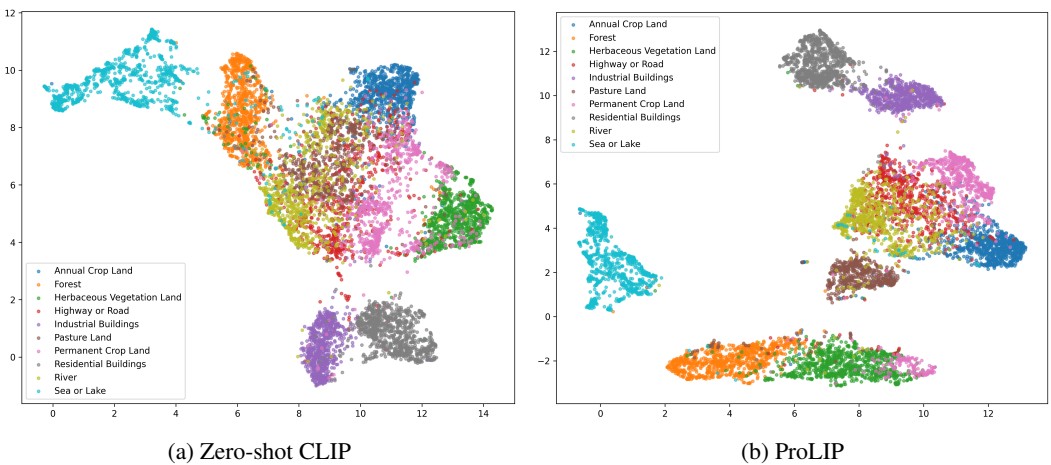

(a) Zero-shot CLIP                      (b) ProLIP

Figure 4: **UMAP Visualization of learned feature manifolds.** We report zero-shot CLIP *vs* and ProLIP on EuroSAT, showing that some classes (e.g.,, 'Pasture Land', 'Permanent Crop Land', 'Sea or Lake', etc.) are better clustered with our method.

Table 6: **Comparison to state-of-the-art methods**. Average classification accuracy (%) and standard deviation over 10 tasks for 11 benchmarks. Best values are highlighted in bold.

| Dataset | Method | $N = 1$ | 2 | 4 | 8 | 16 |
|---------|--------|---------|---|---|---|----|
| | Zero-shot CLIP (Radford et al., 2021) | | | 60.35 | | |
| *ImageNet* | CoOp (Zhou et al., 2022b) | $61.19 \pm 0.17$ | $61.58 \pm 0.40$ | $62.22 \pm 0.22$ | $62.87 \pm 0.21$ | $63.70 \pm 0.13$ |
| | PLOT (Chen et al., 2023) | $60.46 \pm 0.34$ | $60.73 \pm 0.60$ | $61.79 \pm 0.39$ | $62.48 \pm 0.32$ | $63.08 \pm 0.26$ |
| | KgCoOp (Yao et al., 2023) | $60.90 \pm 0.16$ | $61.44 \pm 0.15$ | $62.00 \pm 0.11$ | $62.20 \pm 0.15$ | $62.43 \pm 0.11$ |
| | ProGrad (Zhu et al., 2023) | $\mathbf{61.58} \pm 0.27$ | $\mathbf{62.14} \pm 0.13$ | $\mathbf{62.59} \pm 0.09$ | $63.04 \pm 0.11$ | $63.54 \pm 0.18$ |
| | CLIP-Adapter (Gao et al., 2024) | $59.82 \pm 0.11$ | $59.94 \pm 0.05$ | $59.97 \pm 0.04$ | $59.98 \pm 0.09$ | $61.31 \pm 0.39$ |
| | Tip-Adapter-F (Zhang et al., 2022) | $60.59 \pm 0.14$ | $61.42 \pm 0.05$ | $62.12 \pm 0.06$ | $63.41 \pm 0.07$ | $\mathbf{65.06} \pm 0.04$ |
| | Tip-Adapter-F* (Zhang et al., 2022) | $60.98 \pm 0.15$ | $61.23 \pm 0.11$ | $61.72 \pm 0.25$ | $62.84 \pm 0.10$ | $64.03 \pm 0.12$ |
| | Standard LP (Radford et al., 2021) | $22.21 \pm 0.31$ | $31.96 \pm 0.25$ | $41.48 \pm 0.25$ | $49.49 \pm 0.16$ | $56.04 \pm 0.13$ |
| | LP++ (Huang et al., 2024) | $61.18 \pm 0.08$ | $61.56 \pm 0.14$ | $62.55 \pm 0.12$ | $\mathbf{63.76} \pm 0.07$ | $64.73 \pm 0.05$ |
| | ProLIP | $61.29 \pm 0.13$ | $61.81 \pm 0.18$ | $62.37 \pm 0.18$ | $63.30 \pm 0.11$ | $64.28 \pm 0.12$ |
| | Zero-shot CLIP (Radford et al., 2021) | | | 58.85 | | |
| *SUN397* | CoOp (Zhou et al., 2022b) | $61.79 \pm 0.45$ | $63.32 \pm 0.47$ | $65.79 \pm 0.44$ | $67.89 \pm 0.38$ | $70.15 \pm 0.32$ |
| | PLOT (Chen et al., 2023) | $62.53 \pm 0.30$ | $63.87 \pm 0.26$ | $65.85 \pm 0.48$ | $67.83 \pm 0.36$ | $69.90 \pm 0.31$ |
| | KgCoOp (Yao et al., 2023) | $\mathbf{62.91} \pm 0.59$ | $64.38 \pm 0.30$ | $66.06 \pm 0.37$ | $66.66 \pm 1.10$ | $67.68 \pm 0.78$ |
| | ProGrad (Zhu et al., 2023) | $62.79 \pm 0.50$ | $64.12 \pm 0.55$ | $66.32 \pm 0.59$ | $68.33 \pm 0.28$ | $70.18 \pm 0.27$ |
| | CLIP-Adapter (Gao et al., 2024) | $60.78 \pm 0.16$ | $61.79 \pm 0.23$ | $63.84 \pm 0.35$ | $66.26 \pm 0.14$ | $67.66 \pm 1.05$ |
| | Tip-Adapter-F (Zhang et al., 2022) | $61.02 \pm 0.36$ | $62.15 \pm 0.28$ | $63.86 \pm 0.19$ | $67.25 \pm 0.16$ | $70.94 \pm 0.13$ |
| | Tip-Adapter-F* (Zhang et al., 2022) | $62.58 \pm 0.22$ | $63.79 \pm 0.13$ | $65.49 \pm 0.35$ | $67.43 \pm 0.11$ | $69.25 \pm 0.16$ |
| | Standard LP (Radford et al., 2021) | $32.56 \pm 0.40$ | $43.77 \pm 0.41$ | $54.49 \pm 0.39$ | $61.83 \pm 0.30$ | $67.03 \pm 0.16$ |
| | LP++ (Huang et al., 2024) | $62.47 \pm 0.27$ | $64.65 \pm 0.25$ | $67.28 \pm 0.27$ | $\mathbf{69.34} \pm 0.14$ | $71.23 \pm 0.07$ |
| | ProLIP | $62.71 \pm 0.46$ | $\mathbf{65.58} \pm 0.13$ | $\mathbf{67.68} \pm 0.46$ | $69.17 \pm 0.07$ | $\mathbf{71.29} \pm 0.23$ |
| | Zero-shot CLIP (Radford et al., 2021) | | | 42.69 | | |
| *DTD* | CoOp (Zhou et al., 2022b) | $42.31 \pm 1.86$ | $47.13 \pm 1.93$ | $54.06 \pm 1.49$ | $59.21 \pm 0.91$ | $63.67 \pm 0.83$ |
| | PLOT (Chen et al., 2023) | $45.82 \pm 1.72$ | $51.32 \pm 1.61$ | $55.67 \pm 1.14$ | $61.38 \pm 1.04$ | $65.29 \pm 1.05$ |
| | KgCoOp (Yao et al., 2023) | $45.46 \pm 2.83$ | $50.01 \pm 2.71$ | $53.37 \pm 0.71$ | $58.38 \pm 1.34$ | $62.71 \pm 0.92$ |
| | ProGrad (Zhu et al., 2023) | $44.19 \pm 2.38$ | $50.41 \pm 1.74$ | $54.82 \pm 1.28$ | $60.31 \pm 0.99$ | $63.89 \pm 0.88$ |
| | CLIP-Adapter (Gao et al., 2024) | $43.49 \pm 0.68$ | $44.49 \pm 1.07$ | $48.95 \pm 0.85$ | $57.52 \pm 0.67$ | $62.97 \pm 0.60$ |
| | Tip-Adapter-F (Zhang et al., 2022) | $46.92 \pm 1.01$ | $48.50 \pm 1.08$ | $57.16 \pm 0.53$ | $62.38 \pm 0.47$ | $65.23 \pm 0.82$ |
| | Tip-Adapter-F* (Zhang et al., 2022) | $47.68 \pm 1.43$ | $52.24 \pm 0.74$ | $56.09 \pm 0.99$ | $61.05 \pm 0.71$ | $65.04 \pm 0.21$ |
| | Standard LP (Radford et al., 2021) | $29.63 \pm 1.53$ | $41.19 \pm 1.45$ | $51.72 \pm 0.57$ | $58.78 \pm 0.52$ | $64.56 \pm 0.69$ |
| | LP++ (Huang et al., 2024) | $46.97 \pm 1.37$ | $52.44 \pm 0.99$ | $57.75 \pm 0.82$ | $62.42 \pm 0.53$ | $66.40 \pm 0.50$ |
| | ProLIP | $\mathbf{49.99} \pm 2.28$ | $\mathbf{54.93} \pm 1.29$ | $\mathbf{59.25} \pm 1.03$ | $\mathbf{64.12} \pm 0.64$ | $\mathbf{67.69} \pm 0.87$ |
| | Zero-shot CLIP (Radford et al., 2021) | | | 85.84 | | |
| *Caltech101* | CoOp (Zhou et al., 2022b) | $87.06 \pm 1.24$ | $89.14 \pm 0.87$ | $90.00 \pm 0.63$ | $91.00 \pm 0.66$ | $91.77 \pm 0.29$ |
| | PLOT (Chen et al., 2023) | $\mathbf{89.41} \pm 0.41$ | $\mathbf{90.22} \pm 0.25$ | $90.69 \pm 0.37$ | $91.55 \pm 0.38$ | $92.17 \pm 0.30$ |
| | KgCoOp (Yao et al., 2023) | $88.24 \pm 0.49$ | $88.85 \pm 0.43$ | $89.89 \pm 0.31$ | $90.32 \pm 0.43$ | $90.93 \pm 0.26$ |
| | ProGrad (Zhu et al., 2023) | $88.34 \pm 1.64$ | $89.01 \pm 0.61$ | $90.13 \pm 0.45$ | $90.76 \pm 0.32$ | $91.67 \pm 0.39$ |
| | CLIP-Adapter (Gao et al., 2024) | $87.69 \pm 0.41$ | $89.37 \pm 0.29$ | $90.21 \pm 0.32$ | $91.33 \pm 0.15$ | $92.10 \pm 0.20$ |
| | Tip-Adapter-F (Zhang et al., 2022) | $87.35 \pm 0.64$ | $88.17 \pm 0.29$ | $89.49 \pm 0.25$ | $90.54 \pm 0.34$ | $92.10 \pm 0.25$ |
| | Tip-Adapter-F* (Zhang et al., 2022) | $88.68 \pm 0.44$ | $89.36 \pm 0.59$ | $90.40 \pm 0.26$ | $91.62 \pm 0.23$ | $92.63 \pm 0.21$ |
| | Standard LP (Radford et al., 2021) | $68.88 \pm 1.68$ | $78.41 \pm 0.54$ | $84.91 \pm 0.45$ | $88.70 \pm 0.40$ | $91.14 \pm 0.19$ |
| | LP++ (Huang et al., 2024) | $88.56 \pm 0.43$ | $89.53 \pm 0.35$ | $90.87 \pm 0.19$ | $91.84 \pm 0.24$ | $92.73 \pm 0.17$ |
| | ProLIP | $89.16 \pm 0.48$ | $89.48 \pm 1.15$ | $\mathbf{91.44} \pm 0.42$ | $\mathbf{92.43} \pm 0.32$ | $\mathbf{93.39} \pm 0.38$ |
| | Zero-shot CLIP (Radford et al., 2021) | | | 61.80 | | |
| *UCF101* | CoOp (Zhou et al., 2022b) | $62.80 \pm 1.26$ | $65.62 \pm 1.09$ | $68.69 \pm 0.76$ | $72.57 \pm 0.80$ | $76.39 \pm 0.54$ |
| | PLOT (Chen et al., 2023) | $63.22 \pm 1.05$ | $66.49 \pm 0.92$ | $70.12 \pm 0.62$ | $74.63 \pm 0.79$ | $77.39 \pm 0.53$ |
| | KgCoOp (Yao et al., 2023) | $64.37 \pm 1.66$ | $64.91 \pm 1.01$ | $68.41 \pm 0.38$ | $69.86 \pm 0.33$ | $71.73 \pm 0.78$ |
| | ProGrad (Zhu et al., 2023) | $65.13 \pm 0.87$ | $66.57 \pm 0.62$ | $69.80 \pm 0.62$ | $73.01 \pm 0.52$ | $75.76 \pm 0.47$ |
| | CLIP-Adapter (Gao et al., 2024) | $64.25 \pm 0.54$ | $66.68 \pm 0.31$ | $69.77 \pm 0.40$ | $73.90 \pm 0.50$ | $77.26 \pm 0.39$ |
| | Tip-Adapter-F (Zhang et al., 2022) | $64.28 \pm 0.54$ | $65.48 \pm 0.43$ | $67.61 \pm 0.28$ | $72.05 \pm 0.53$ | $77.30 \pm 0.21$ |
| | Tip-Adapter-F* (Zhang et al., 2022) | $65.50 \pm 0.59$ | $68.55 \pm 0.45$ | $70.55 \pm 0.58$ | $74.25 \pm 0.48$ | $76.83 \pm 0.24$ |
| | Standard LP (Radford et al., 2021) | $40.80 \pm 1.05$ | $51.71 \pm 0.79$ | $61.64 \pm 0.50$ | $68.47 \pm 0.44$ | $73.38 \pm 0.43$ |
| | LP++ (Huang et al., 2024) | $65.41 \pm 0.37$ | $69.20 \pm 0.52$ | $71.68 \pm 0.41$ | $74.86 \pm 0.36$ | $77.46 \pm 0.39$ |
| | ProLIP | $\mathbf{67.05} \pm 0.12$ | $\mathbf{70.02} \pm 0.46$ | $\mathbf{71.90} \pm 1.01$ | $\mathbf{76.36} \pm 0.67$ | $\mathbf{80.29} \pm 0.22$ |
| | Zero-shot CLIP (Radford et al., 2021) | | | 65.98 | | |
| *Flowers102* | CoOp (Zhou et al., 2022b) | $69.00 \pm 2.44$ | $78.47 \pm 1.88$ | $85.34 \pm 1.69$ | $91.68 \pm 0.82$ | $94.47 \pm 0.36$ |
| | PLOT (Chen et al., 2023) | $71.09 \pm 1.44$ | $81.22 \pm 0.92$ | $87.61 \pm 0.79$ | $92.60 \pm 0.55$ | $\mathbf{95.18} \pm 0.40$ |
| | KgCoOp (Yao et al., 2023) | $68.73 \pm 1.97$ | $69.63 \pm 1.25$ | $76.51 \pm 0.51$ | $80.71 \pm 0.63$ | $84.48 \pm 0.70$ |
| | ProGrad (Zhu et al., 2023) | $72.16 \pm 1.74$ | $79.55 \pm 0.88$ | $84.56 \pm 1.41$ | $91.73 \pm 0.35$ | $94.10 \pm 0.41$ |
| | CLIP-Adapter (Gao et al., 2024) | $66.86 \pm 0.73$ | $69.71 \pm 0.46$ | $77.42 \pm 0.60$ | $87.20 \pm 0.52$ | $91.16 \pm 0.23$ |
| | Tip-Adapter-F (Zhang et al., 2022) | $67.73 \pm 0.57$ | $68.18 \pm 0.84$ | $71.17 \pm 0.67$ | $84.11 \pm 0.49$ | $93.02 \pm 0.28$ |
| | Tip-Adapter-F* (Zhang et al., 2022) | $\mathbf{78.46} \pm 1.01$ | $\mathbf{85.14} \pm 0.81$ | $88.53 \pm 0.54$ | $92.33 \pm 0.32$ | $94.26 \pm 0.38$ |
| | Standard LP (Radford et al., 2021) | $56.98 \pm 1.56$ | $73.40 \pm 0.87$ | $84.38 \pm 0.53$ | $91.81 \pm 0.34$ | $95.05 \pm 0.29$ |
| | LP++ (Huang et al., 2024) | $78.21 \pm 1.01$ | $84.69 \pm 0.70$ | $\mathbf{89.56} \pm 0.45$ | $92.61 \pm 0.32$ | $94.26 \pm 0.24$ |
| | ProLIP | $76.13 \pm 1.35$ | $82.31 \pm 1.36$ | $88.37 \pm 0.78$ | $\mathbf{92.79} \pm 0.61$ | $95.15 \pm 0.35$ |

Table 7: **Comparison to state-of-the-art methods** (Continued). Average classification accuracy (%) and standard deviation over 10 tasks for 11 benchmarks. Best values are highlighted in bold.

| Dataset | Method | $N = 1$ | 2 | 4 | 8 | 16 |
|---|---|---|---|---|---|---|
| | Zero-shot CLIP (Radford et al., 2021) | | | 55.78 | | |
| | CoOp (Zhou et al., 2022b) | $57.00 \pm 0.93$ | $58.96 \pm 0.78$ | $62.81 \pm 0.71$ | $68.40 \pm 0.61$ | $72.87 \pm 0.50$ |
| | PLOT (Chen et al., 2023) | $57.47 \pm 0.58$ | $59.89 \pm 0.60$ | $63.49 \pm 0.80$ | $68.75 \pm 0.46$ | $73.86 \pm 0.39$ |
| | KgCoOp (Yao et al., 2023) | $57.19 \pm 0.65$ | $58.94 \pm 0.33$ | $59.85 \pm 0.51$ | $61.42 \pm 0.40$ | $62.99 \pm 1.33$ |
| *StanfordCars* | ProGrad (Zhu et al., 2023) | $\mathbf{58.63} \pm 0.39$ | $61.23 \pm 0.65$ | $65.02 \pm 0.78$ | $69.43 \pm 0.40$ | $72.76 \pm 0.45$ |
| | CLIP-Adapter (Gao et al., 2024) | $56.67 \pm 0.22$ | $57.94 \pm 0.27$ | $61.13 \pm 0.30$ | $65.43 \pm 0.10$ | $70.24 \pm 0.79$ |
| | Tip-Adapter-F (Zhang et al., 2022) | $57.24 \pm 0.23$ | $58.12 \pm 0.50$ | $59.34 \pm 0.20$ | $64.25 \pm 0.19$ | $71.38 \pm 0.23$ |
| | Tip-Adapter-F* (Zhang et al., 2022) | $57.85 \pm 0.33$ | $60.55 \pm 0.34$ | $64.22 \pm 0.52$ | $68.75 \pm 0.31$ | $74.19 \pm 0.30$ |
| | Standard LP (Radford et al., 2021) | $22.94 \pm 0.61$ | $35.48 \pm 0.51$ | $47.49 \pm 0.67$ | $59.34 \pm 0.30$ | $69.11 \pm 0.18$ |
| | LP++ (Huang et al., 2024) | $57.20 \pm 0.65$ | $59.95 \pm 0.36$ | $63.44 \pm 0.34$ | $67.81 \pm 0.24$ | $72.33 \pm 0.18$ |
| | ProLIP | $58.29 \pm 0.49$ | $\mathbf{61.94} \pm 0.37$ | $\mathbf{66.09} \pm 0.27$ | $\mathbf{69.82} \pm 0.43$ | $\mathbf{76.03} \pm 0.09$ |
| | Zero-shot CLIP (Radford et al., 2021) | | | 17.07 | | |
| | CoOp (Zhou et al., 2022b) | $12.50 \pm 6.16$ | $17.59 \pm 3.70$ | $21.27 \pm 2.54$ | $26.85 \pm 0.63$ | $31.20 \pm 0.40$ |
| | PLOT (Chen et al., 2023) | $17.75 \pm 1.36$ | $19.55 \pm 0.99$ | $22.26 \pm 0.89$ | $26.70 \pm 0.70$ | $32.09 \pm 0.68$ |
| | KgCoOp (Yao et al., 2023) | $18.61 \pm 0.76$ | $18.93 \pm 1.01$ | $21.16 \pm 0.82$ | $22.80 \pm 0.44$ | $24.10 \pm 0.59$ |
| *FGVCAircraft* | ProGrad (Zhu et al., 2023) | $18.41 \pm 0.98$ | $20.51 \pm 1.11$ | $23.65 \pm 0.50$ | $26.98 \pm 0.50$ | $30.47 \pm 0.76$ |
| | CLIP-Adapter (Gao et al., 2024) | $18.56 \pm 0.20$ | $19.18 \pm 0.28$ | $21.00 \pm 0.21$ | $23.76 \pm 0.33$ | $33.37 \pm 0.23$ |
| | Tip-Adapter-F (Zhang et al., 2022) | $18.23 \pm 0.19$ | $19.12 \pm 0.20$ | $20.55 \pm 0.20$ | $23.60 \pm 0.29$ | $30.37 \pm 0.25$ |
| | Tip-Adapter-F* (Zhang et al., 2022) | $19.08 \pm 0.15$ | $20.79 \pm 0.59$ | $23.99 \pm 0.57$ | $30.58 \pm 0.29$ | $36.16 \pm 0.34$ |
| | Standard LP (Radford et al., 2021) | $12.66 \pm 0.59$ | $16.92 \pm 0.56$ | $21.11 \pm 0.83$ | $26.53 \pm 0.38$ | $32.42 \pm 0.54$ |
| | LP++ (Huang et al., 2024) | $\mathbf{19.69} \pm 0.39$ | $\mathbf{21.58} \pm 0.46$ | $24.22 \pm 0.60$ | $27.73 \pm 0.48$ | $31.73 \pm 0.44$ |
| | ProLIP | $17.86 \pm 1.18$ | $20.88 \pm 0.69$ | $\mathbf{27.35} \pm 0.19$ | $\mathbf{32.59} \pm 0.37$ | $\mathbf{40.09} \pm 0.12$ |
| | Zero-shot CLIP (Radford et al., 2021) | | | 36.22 | | |
| | CoOp (Zhou et al., 2022b) | $40.36 \pm 7.19$ | $56.15 \pm 5.82$ | $66.13 \pm 3.62$ | $77.02 \pm 1.78$ | $82.59 \pm 1.00$ |
| | PLOT (Chen et al., 2023) | $44.22 \pm 9.14$ | $64.19 \pm 6.24$ | $69.37 \pm 3.26$ | $78.84 \pm 1.33$ | $81.76 \pm 1.43$ |
| | KgCoOp (Yao et al., 2023) | $43.86 \pm 9.17$ | $52.92 \pm 5.92$ | $59.51 \pm 3.46$ | $63.23 \pm 3.03$ | $64.04 \pm 1.40$ |
| *EuroSAT* | ProGrad (Zhu et al., 2023) | $49.37 \pm 5.03$ | $65.22 \pm 4.01$ | $69.57 \pm 2.88$ | $78.44 \pm 1.69$ | $82.17 \pm 0.98$ |
| | CLIP-Adapter (Gao et al., 2024) | $43.00 \pm 2.27$ | $48.60 \pm 2.76$ | $59.15 \pm 2.26$ | $69.92 \pm 1.49$ | $75.38 \pm 0.78$ |
| | Tip-Adapter-F (Zhang et al., 2022) | $47.63 \pm 2.64$ | $57.62 \pm 1.86$ | $69.30 \pm 2.41$ | $75.22 \pm 1.32$ | $78.59 \pm 1.48$ |
| | Tip-Adapter-F* (Zhang et al., 2022) | $49.27 \pm 2.88$ | $65.66 \pm 1.39$ | $70.72 \pm 2.73$ | $74.66 \pm 3.15$ | $78.73 \pm 0.81$ |
| | Standard LP (Radford et al., 2021) | $48.29 \pm 2.95$ | $56.81 \pm 2.93$ | $64.99 \pm 3.47$ | $74.56 \pm 0.98$ | $80.29 \pm 0.90$ |
| | LP++ (Huang et al., 2024) | $57.23 \pm 1.63$ | $61.65 \pm 1.66$ | $68.67 \pm 2.21$ | $75.86 \pm 0.99$ | $80.53 \pm 1.05$ |
| | ProLIP | $\mathbf{65.78} \pm 0.45$ | $\mathbf{67.26} \pm 0.57$ | $\mathbf{77.03} \pm 1.19$ | $\mathbf{80.08} \pm 0.48$ | $\mathbf{85.82} \pm 0.63$ |
| | Zero-shot CLIP (Radford et al., 2021) | | | 85.75 | | |
| | CoOp (Zhou et al., 2022b) | $86.27 \pm 1.17$ | $86.33 \pm 1.13$ | $85.34 \pm 1.69$ | $87.85 \pm 1.21$ | $88.68 \pm 0.71$ |
| | PLOT (Chen et al., 2023) | $87.15 \pm 0.72$ | $87.23 \pm 1.21$ | $88.03 \pm 0.49$ | $88.38 \pm 0.64$ | $88.23 \pm 0.54$ |
| | KgCoOp (Yao et al., 2023) | $87.51 \pm 0.68$ | $87.51 \pm 0.75$ | $88.04 \pm 0.46$ | $88.59 \pm 0.34$ | $89.28 \pm 0.21$ |
| *OxfordPets* | ProGrad (Zhu et al., 2023) | $\mathbf{88.34} \pm 0.65$ | $\mathbf{87.88} \pm 0.69$ | $\mathbf{88.59} \pm 0.58$ | $\mathbf{88.87} \pm 0.42$ | $\mathbf{89.39} \pm 0.47$ |
| | CLIP-Adapter (Gao et al., 2024) | $85.46 \pm 0.48$ | $86.37 \pm 0.25$ | $87.21 \pm 0.51$ | $87.95 \pm 0.26$ | $88.33 \pm 0.33$ |
| | Tip-Adapter-F (Zhang et al., 2022) | $85.70 \pm 0.16$ | $86.05 \pm 0.46$ | $86.40 \pm 0.29$ | $87.66 \pm 0.28$ | $89.08 \pm 0.27$ |
| | Tip-Adapter-F* (Zhang et al., 2022) | $86.05 \pm 0.36$ | $86.49 \pm 0.61$ | $87.19 \pm 0.36$ | $87.89 \pm 0.34$ | $88.26 \pm 0.37$ |
| | Standard LP (Radford et al., 2021) | $30.62 \pm 1.61$ | $42.64 \pm 2.03$ | $55.60 \pm 0.98$ | $67.32 \pm 0.98$ | $76.23 \pm 0.38$ |
| | LP++ (Huang et al., 2024) | $84.24 \pm 1.36$ | $85.74 \pm 0.56$ | $86.94 \pm 0.48$ | $87.71 \pm 0.65$ | $88.38 \pm 0.61$ |
| | ProLIP | $85.62 \pm 1.10$ | $86.02 \pm 1.50$ | $87.24 \pm 0.75$ | $88.20 \pm 0.56$ | $89.00 \pm 0.51$ |
| | Zero-shot CLIP (Radford et al., 2021) | | | 77.35 | | |
| | CoOp (Zhou et al., 2022b) | $75.58 \pm 1.29$ | $77.49 \pm 0.41$ | $77.93 \pm 0.58$ | $78.92 \pm 0.19$ | $79.21 \pm 0.36$ |
| | PLOT (Chen et al., 2023) | $77.46 \pm 0.55$ | $77.72 \pm 0.26$ | $78.23 \pm 0.25$ | $78.40 \pm 0.35$ | $78.86 \pm 0.19$ |
| | KgCoOp (Yao et al., 2023) | $77.20 \pm 0.77$ | $\mathbf{78.04} \pm 0.18$ | $77.97 \pm 0.28$ | $78.39 \pm 0.40$ | $78.73 \pm 0.23$ |
| *Food101* | ProGrad (Zhu et al., 2023) | $\mathbf{78.36} \pm 0.41$ | $78.01 \pm 0.70$ | $\mathbf{78.38} \pm 0.87$ | $\mathbf{79.11} \pm 0.18$ | $\mathbf{79.51} \pm 0.23$ |
| | CLIP-Adapter (Gao et al., 2024) | $76.93 \pm 0.19$ | $77.22 \pm 0.15$ | $77.64 \pm 0.17$ | $77.97 \pm 0.22$ | $78.45 \pm 0.14$ |
| | Tip-Adapter-F (Zhang et al., 2022) | $77.53 \pm 0.14$ | $77.53 \pm 0.22$ | $77.82 \pm 0.27$ | $78.26 \pm 0.22$ | $78.99 \pm 0.15$ |
| | Tip-Adapter-F* (Zhang et al., 2022) | $77.58 \pm 0.10$ | $77.36 \pm 0.39$ | $77.78 \pm 0.15$ | $78.17 \pm 0.11$ | $78.72 \pm 0.06$ |
| | Standard LP (Radford et al., 2021) | $31.59 \pm 1.20$ | $44.60 \pm 1.03$ | $56.13 \pm 0.63$ | $64.45 \pm 0.55$ | $70.97 \pm 0.19$ |
| | LP++ (Huang et al., 2024) | $76.61 \pm 0.77$ | $77.22 \pm 0.55$ | $77.79 \pm 0.34$ | $78.53 \pm 0.14$ | $78.88 \pm 0.19$ |
| | ProLIP | $76.60 \pm 0.11$ | $77.71 \pm 0.08$ | $77.15 \pm 0.30$ | $78.57 \pm 0.07$ | $79.02 \pm 0.13$ |

Table 8: **ProLIP sensitivity to hyperparameter choice.** Accuracy of ProLIP to the hyperparameters (learning rate LR and regularization weight $\lambda$) for $N \in \{1, 2, 4, 8, 16\}$-shot settings. Each number is an average over 11 datasets, 10 runs for each.

| Method | | $N = 1$ | 2 | 4 | 8 | 16 |
|---|---|---|---|---|---|---|
| Zero-shot CLIP | | | | 58.89 | | |
| ProLIP (grid search) | | 64.59±0.98 | 67.09±0.87 | **70.53**±0.69 | **73.40**±0.45 | **76.55**±0.41 |
| ProLIP, LR=$10^{-6}$ | $\lambda = 1$ | 63.05±0.52 | 65.00±0.40 | 66.91±0.30 | 68.13±0.31 | 68.95±0.18 |
| | $\lambda = 10^{-1}$ | 63.90±0.60 | 66.32±0.41 | 68.99±0.34 | 70.96±0.31 | 72.50±0.29 |
| | $\lambda = 10^{-2}$ | 63.94±0.65 | 66.46±0.42 | 69.17±0.37 | 71.30±0.33 | 72.89±0.30 |
| | $\lambda = 0$ | 63.94±0.66 | 66.46±0.42 | 69.19±0.37 | 71.34±0.34 | 72.94±0.30 |
| ProLIP, LR=$10^{-5}$ | $\lambda = 1$ | 64.67±0.63 | 66.37±0.35 | 68.34±0.33 | 69.63±0.30 | 70.52±0.21 |
| | $\lambda = 10^{-1}$ | **64.82**±0.71 | 67.13±0.45 | 70.23±0.42 | 72.57±0.33 | 74.88±0.29 |
| | $\lambda = 10^{-2}$ | 63.64±0.86 | 66.47±0.63 | 70.07±0.50 | 72.79±0.32 | 75.77±0.29 |
| | $\lambda = 0$ | 62.91±0.85 | 65.97±0.67 | 69.76±0.55 | 72.57±0.35 | 75.73±0.30 |
| ProLIP, LR=$10^{-4}$ | $\lambda = 1$ | 64.78±0.67 | 66.63±0.41 | 68.88±0.34 | 70.33±0.31 | 71.40±0.26 |
| | $\lambda = 10^{-1}$ | 64.65±0.86 | **67.22**±0.66 | 70.44±0.39 | 72.91±0.31 | 75.34±0.36 |
| | $\lambda = 10^{-2}$ | 60.54±1.47 | 64.29±1.16 | 69.10±0.70 | 72.51±0.40 | 75.88±0.35 |
| | $\lambda = 0$ | 50.61±1.60 | 58.36±1.05 | 65.35±0.67 | 69.70±0.45 | 73.29±0.47 |
| ProLIP, LR=$10^{-3}$ | $\lambda = 1$ | 64.75±0.68 | 66.56±0.46 | 68.89±0.35 | 70.35±0.30 | 71.49±0.25 |
| | $\lambda = 10^{-1}$ | 64.32±0.84 | 66.97±0.59 | 70.39±0.47 | 72.92±0.35 | 75.49±0.37 |
| | $\lambda = 10^{-2}$ | 58.58±2.04 | 63.99±0.97 | 69.15±0.69 | 72.69±0.42 | 76.05±0.38 |
| | $\lambda = 0$ | 40.17±1.65 | 49.08±1.19 | 56.34±1.08 | 59.78±0.92 | 61.67±0.96 |
| ProLIP, LR=$10^{-2}$ | $\lambda = 1$ | 64.61±0.72 | 66.53±0.45 | 68.83±0.37 | 70.29±0.31 | 71.41±0.25 |
| | $\lambda = 10^{-1}$ | 63.47±1.68 | 66.72±0.74 | 70.18±0.57 | 72.66±0.59 | 75.06±0.89 |
| | $\lambda = 10^{-2}$ | 53.29±2.19 | 61.00±1.27 | 67.49±0.73 | 71.62±0.54 | 75.20±0.50 |
| | $\lambda = 0$ | 20.02±2.21 | 23.91±2.17 | 28.05±2.43 | 32.02±1.93 | 35.75±1.25 |

Table 9: **Comparison with full fine-tuning.** We report the classification accuracy (%) averaged over 11 datasets and 10 support sets, along with standard deviation, comparing ProLIP to full fine-tuning of the vision encoder.

| Method | $N = 1$ | 2 | 4 | 8 | 16 |
|---|---|---|---|---|---|
| Zero-shot CLIP | | | 58.89 | | |
| Full Fine-tuning | 46.09±6.33 | 51.85±5.32 | 58.06±6.19 | 62.22±1.23 | 67.74±0.68 |
| ProLIP ($\lambda = 0$) | 62.91±0.85 | 65.97±0.67 | 69.76±0.55 | 72.57±0.35 | 75.73±0.30 |
| ProLIP ($\lambda = 1/N$) | **64.67**±0.63 | 66.80±0.39 | 69.73±0.37 | 72.44±0.33 | 75.34±0.31 |
| ProLIP ($\lambda = 1/N^2$) | **64.67**±0.63 | **67.04**±0.42 | 70.27±0.44 | 72.84±0.32 | 75.77±0.30 |

