# OpenReview forum: "Fine-tuning CLIP’s Last Visual Projector: A Few-Shot Cornucopia"
_ICLR.cc/2025/Conference — ICLR 2025 Conference Withdrawn Submission_

### Official Review · Reviewer_S8em · 2024-11-02

**Soundness:** 2
**Presentation:** 3
**Contribution:** 2
**Rating:** 3
**Confidence:** 4

**Summary:**

This paper proposes only to fine-tune the last projection layer of the vision encoder and finds this simple paradigm can yield on-par or better few-shot classification performance.

**Strengths:**

1. The paper is well-written and easy to follow.
2. The proposed method seems to be simple and effective.

**Weaknesses:**

1. I hope to see more in-depth analysis and discussion to indicate the reasons why ProLIP is effective, rather than descriptions like "Perhaps surprisingly." The current version of the paper makes me feel that the proposed method is trivial.

2. The paper seems to show significant performance improvements only in specific datasets (e.g., EuroSAT) or 16-shot scenarios, while the performance improvement is quite limited in other contexts.

3. The paper conducted a grid search for training parameters, which may be the reason for its superiority over other methods. I am curious whether using a similar parameter search for the baseline methods would diminish ProLIP's advantages.

4. The paper lacks performance comparisons with more advanced methods, such as PromptSRC[1] and CoPrompt[2].

[1] Self-regulating Prompts: Foundational Model Adaptation without Forgetting. ICCV, 2023.

[2] Consistency-guided Prompt Learning for Vision-Language Models. ICLR, 2024.

**Questions:**

Please refer to the 'Weaknesses' part.

---

> ### Author Response · Authors · 2024-11-15
> **Answer to reviewer S8em**
>
> ### W1. In-depth analysis and discussion
> - Please refer to answer **W1** of **Reviewer NwVL** on a similar question.
>
> ### W2. Significant performance improvements only in specific datasets
> - We would like to highlight that ProLIP achieves top-performance on few-shot image classification on CLIP on 11 datasets on different few-shot regimes outperforming runner-up methods by up to 2% (Tab. 1), few-shot domain generalization with highest average performance on both in-distribution and out-of-distribution data (Tab. 3), robustness to distribution shift in test-time adaptation without training data (Tab. 5).
> - On cross-dataset generalization (Tab. 4) zero-shot CLIP outperforms on average all fine-tuning strategies, while ProLIP is on average second to it, outperforming all fine-tuning methods.
> - In addition, ProLIP is extremely simple and does not change the original CLIP architecture nor impacts runtime.
>
> ### W3. The paper conducted a grid search for training parameters
> - We do conduct a grid-search for the tuning of the hyper-parameters and for getting insights in the inner-functioning of ProLIP. However we also show that using the regularizer ($\lambda>0$) is better to avoid severe overfitting, and that state-of-the-art results can be still achieved with a fixed learning rate over all the datasets, and a fixed $\lambda$ which can be chosen as a decreasing function of the number N of shots.
> - For the other methods we report fairly the scores obtained in their original settings with the tunings of their authors or as reported in prior works.
>
> ### W4. Missing performance comparison
> - Thank you for the suggested references. Both PromptSRC and CoPrompt leverage learnable visual prompts and are ViT-specific. Additionally, CoPrompt leverages the combination of learnable adapters and prompts. Their experiments focus on base-to-novel classification and domain/dataset generalization of few-shot learning.
> - In the evaluation of ProLIP we have considered multiple different few-shot protocols and methods that are compatible with both ResNet and ViT encoders in line with recent published works in this area.
> - We will consider these works in the update.

---

### Official Review · Reviewer_xgsb · 2024-11-03

**Soundness:** 3
**Presentation:** 3
**Contribution:** 3
**Rating:** 8
**Confidence:** 4

**Summary:**

This paper introduces ProLIP, a method for few-shot adaptation of CLIP. The core idea of ProLIP is to fine-tune only the final visual projection matrix of the vision encoder during training. Additionally, it incorporates L2 regularization to maintain the matrix's proximity to the pretrained version. ProLIP is simple and efficient, demonstrating strong performance relative to other few-shot methods.

**Strengths:**

1.	The approach is simple and efficient. ProLIP does not introduce new training parameters; instead, it fine-tunes a pretrained projection matrix, making it more efficient than prompt learning.
2.	The performance is strong. ProLIP outperforms methods like LP++ and adapter-based approaches.
3.	The presentation is clear and easy to understand.
4.	The experiments are comprehensive. In addition to comparisons with other state-of-the-art methods, the authors investigate the sensitivity to hyperparameter choices and the parametric λ.

**Weaknesses:**

There are no specific drawbacks noticed.

**Questions:**

How does ProLIP compare to LP++ with the proposed regulation. I believe the primary distinction between ProLIP and LP++ lies in ProLIP's imposition of regularization on the projection layer.

---

> ### Author Response · Authors · 2024-11-15
> **Answer to reviewer xgsb**
>
> ### Q1. How does ProLIP compare to LP++ with the proposed regularization?
> - ProLIP fine-tunes the last visual projection matrix of the vision encoder and the regularization keeps these trainable weights close to the pretrained ones. LP++ keeps the entire vision encoder frozen and adds a linear classifier that is initialized from text embeddings. Adding our regularization to LP++ would not help as the encoder is frozen. However, adding regularization to the newly initialized linear classifier could be possible but unclear in impact as LP++ already has specific regularization from its optimizer. We will explore this in future work.

---

### Official Review · Reviewer_BdGM · 2024-11-04

**Soundness:** 2
**Presentation:** 2
**Contribution:** 1
**Rating:** 5
**Confidence:** 4

**Summary:**

This paper introduces ProLIP, a method for few-shot adaptation of the CLIP vision-language model. ProLIP fine-tunes only the final projection layer of CLIP’s visual encoder, eliminating the need for additional parameters or complex architecture changes. This approach aligns with CLIP's training by using text embeddings as classification weights, while a regularization term ensures that the updated weights stay close to their pretrained values, reducing overfitting. Experiments show that ProLIP achieves competitive or superior performance to existing few-shot methods across tasks such as cross-dataset transfer and domain generalization.

**Strengths:**

1: ProLIP requires only fine-tuning of the last projection layer in CLIP’s visual encoder, eliminating the need for additional parameters, such as external adapters or prompt-tuning layers. This reduces memory and computational costs, making it feasible for resource-limited scenarios.

2: The proposed method is simple, i.e., only fine-tuning the visual projector in the CLIP model.

3: ProLIP includes a norm regularization that constrains the projection matrix’s weights, keeping them close to their pre-trained values. This regularization not only prevents overfitting in few-shot scenarios but also enhances the model’s stability and generalization.

**Weaknesses:**

1: The overall work sounds a little bit trivial. This paper mainly claims that only fine-tuning the final projector layer (part of model weights) can benefit the few-shot classification task. The modification is a kind of simple baseline rather than a novel method. There are also fewer provided insights into the design and the scope of downstream tasks is only limited to the few-shot task.

2: Misaligned results. The results of the proposed ProCLIP are not consistent between Table 1 and Table 2.

**Questions:**

N/A.

---

> ### Author Response · Authors · 2024-11-15
> **Answer to reviewer BdGM**
>
> ### W1. Trivial work
> - Our approach is simple and effective. Despite its extreme simplicity, adapting CLIP using our approach has not been proposed in the literature. Please refer to answer **W1** of **Reviewer biKw** for additional responses.
>
> ### W2. Few insights into the design
> - Please refer to answer **W1** of **Reviewer NwVL** on a similar question.
>
> ### W3. Limited scope
> - Indeed we do not make other claims than few-shot learning for vision-language models, as also illustrated in the title. We believe that in the era of large foundation models trained with minimal or no supervision on large amounts of data, devising strategies to adapt such models on few labeled samples with good performance and without overfitting is a challenging and important research topic and application. This is also confirmed by the numerous publications in this area in the past 2 years.
> - Besides, in the evaluation we consider multiple types of experiments and settings: {1,2,4,8,16}-shot learning on 11 datasets, domain generalization, cross-dataset generalization, test-time adaptation.
>
>
> ### W4. Misaligned results in Tab. 1 and 2
> - We believe there is a misunderstanding from the reviewer here. In Tab. 1 (Sec. 3.1) we report ProLIP results with hyper-parameters identified by grid-search and compare with SoTA methods. In Tab. 2 (Sec. 3.2) we report results for our strategy to set the regularization $\lambda$ as a function of N with different learning rate values.
> - The aim of Tab. 2 is to show that we can obtain good (SoTA) scores for ProLIP without much effort spent on hyper-parameter tuning.

---

### Official Review · Reviewer_biKw · 2024-11-04

**Soundness:** 2
**Presentation:** 2
**Contribution:** 2
**Rating:** 3
**Confidence:** 4

**Summary:**

The paper introduces ProLIP,  which involves fine-tuning only the last visual projection matrix of the CLIP model, for post-training. This fine-tuning uses a masked reconstruction loss to learn semantic contributions for each image patch, enhancing the model's ability to capture local semantics without the need for additional annotated data.

**Strengths:**

1. The paper evaluates ProLIP on various vision-centric and vision-language benchmarks. Besides, the method is shown to be effective across different datasets in few-shot classification, domain generalization, cross-dataset transfer, and test-time adaptation.
2. The method is applicable to various models trained with image-level supervision, including CLIP and SigLIP, and can be used for different vision-language tasks.

**Weaknesses:**

1. The main concern is the novelty, the approach presented in this article is more of a trivial trick than a novel academic contribution. To the best of my knowledge, fine-tuning the mlp of the last layer of a large model is a very common trivial trick for fine-tuning either large language models and mllm models.
2. The experimental performance still falls short of the state-of-the-art approachs [1,2,3,4], suggesting that such a simple strategy is not the best one for fine-tuning.
3. The performance of ProLIP is sensitive to the choice of regularization strength ($\lambda$), which may require tuning or adaptive strategies for different datasets or tasks.
4. The paper assumes that pre-trained models contain sufficient knowledge for local semantics, which may not always be the case and could limit the method's effectiveness in certain scenarios.

[1] Khattak, Muhammad Uzair, et al. "Maple: Multi-modal prompt learning." Proceedings of the IEEE/CVF Conference on Computer Vision and Pattern Recognition. 2023.
[2] Khattak, Muhammad Uzair, et al. "Self-regulating prompts: Foundational model adaptation without forgetting." Proceedings of the IEEE/CVF International Conference on Computer Vision. 2023.
[3] Wang, Yaoming, et al. "VioLET: Vision-Language Efficient Tuning with Collaborative Multi-modal Gradients." Proceedings of the 31st ACM International Conference on Multimedia. 2023.
[4] Roy, Shuvendu, and Ali Etemad. "Consistency-guided prompt learning for vision-language models." arXiv preprint arXiv:2306.01195 (2023).

**Questions:**

Please refer to the weaknesses.

---

> ### Author Response · Authors · 2024-11-15
> **Answer to reviewer biKw**
>
> ### S1. Potentially mistaken review summary
> > This fine-tuning uses a masked reconstruction loss to learn semantic contributions for each image patch, enhancing the model's ability to capture local semantics without the need for additional annotated data.
> - This summary does not seem to describe our method for few-shot fine-tuning of CLIP for image classification. We do not use masking, patch-level semantics or reconstruction loss.
>
>
> ### W1. Novelty - trivial trick
> - Our approach is indeed very simple and highly practical, but we do not consider it as a trivial trick. To the best of our knowledge such an approach has not been proposed before in the CLIP adaptation literature.
> - Most of the related approaches add new layers or adapters or learnable prompts on the vision encoder or text encoder side. However they change the original architecture and/or require full back-propagation steps, impacting both training and inference time.
> - Linear probing variants add a new layer on top of CLIP and train with a regular classification loss.
> - ProLIP does not change the architecture of CLIP and fine-tunes only the last vision projection layer. The supervision signal is given via text embeddings, thus remaining faithful to the original pretraining regime of CLIP. In addition we propose a regularization strategy to prevent overfitting during the few-shot training.
> -  Moreover, the architecture-agnostic nature of ProLIP makes it generic and suitable to different multi-modal pretrained networks (ResNet, ViT). Due to intrinsic differences across architectures, finding a unified method to efficiently fine-tune vision-language models based on the pretrained model weights is not trivial.
>
>
> ### W2. Experimental performance
> - Thank you for the suggested references. In our evaluation we have considered different few-shot protocols and methods that are compatible with both ResNet and ViT encoders in line with recent published works in this area.
> - MaPLe is a ViT-specific method as it uses learnable visual prompts. It focuses on base-to-novel classification and domain/dataset generalization of 16-shot learning. Preliminary experiments on base-to-novel generalization with ViT-B/16 show that ProLIP is better than MaPLe on base classes and worse on novel, with similar harmonic mean scores.
> - We will update the references and experiments.
>
> ### W3. Sensitivity to choice of regularization strength $\lambda$
> - We respectfully disagree. We show that the use of different regularization strengths reduces overfitting and it is compatible with a wide range of learning rate values (See Fig.2 and Tab. 8). When no regularization is applied, accuracy drops dramatically for learning rates higher than 1e-5.
> - Besides we propose in Sec. 3.2 a strategy to reliably set $\lambda$ as a function of the number of shots $N$ available for training (see Tab. 2)
>
> ### W4. The paper assumes that pre-trained models contain sufficient knowledge for local semantics
> - ProLIP does not make any assumptions on local semantics. ProLIP deals with image classification and fine-tunes the last projection layer of CLIP that processes global features.

---

### Official Review · Reviewer_NwVL · 2024-11-05

**Soundness:** 3
**Presentation:** 3
**Contribution:** 2
**Rating:** 5
**Confidence:** 3

**Summary:**

This paper proposes ProLIP, a parameter-efficient method for adapting CLIP to few-shot classification. ProLIP proposes to finetune only the last projection layer of the vision encoder without adding other parameters or requiring prompt engineering. It also employs regularization of the weight matrix to preserve pretrained knowledge of the original CLIP. It achieves strong performance on few-shot classification and test-time adaptation.

**Strengths:**

* ProLIP provides a very simple and efficient adaptation compared to the previous methods, by finetuning only the last projection layer of the vision encoder.

* ProLIP shows comparable or superior performance on various tasks from few-shot classification, domain generalization and test-time adaptation.

**Weaknesses:**

There are several aspects where the paper could be improved:

* A more in-depth analysis of how ProLIP achieves competitive results compared to more parameter-intensive methods. This would help providing insights into why this simple approach is so effective.

* Experiments with larger backbones, like ViT-Large, could be beneficial. Given that the method finetunes only the last projection layer, model scalability may be one of ProLIP’s strengths and could add depth to this.

* Additional evaluations on diverse out-of-domain datasets. Since ProLIP fine-tunes only a very small portion of the model, it would be interesting to see the range of new data domains and tasks that ProLIP can adapt (or where it fails) to in few-shot settings.

* Ablation on different shot numbers. It would be beneficial to have additional analysis of ProLIP’s performance across different few-shot settings (e.g., from 1-shot to N-shots), to show how well ProLIP leverages an increasing number of shots.

* ProLIP could potentially struggle on tasks with longer texts as the text encoder is kept completely frozen, compared to the prompt tuning methods.

**Questions:**

* Could ProLIP be combined with prompt tuning methods, and if so, would this joint approach yield even greater performance improvements? Exploring this combination might reveal synergistic benefits of ProLIP’s efficient layer tuning with prompt-based adaptations.

* How does ProLIP perform on datasets not represented in CLIP’s pretraining (e.g., medical or aerial imagery)? Expanding on the cross-dataset experiments could further demonstrate ProLIP’s adaptability to diverse data distributions.

* Comparison of training time and computation against prompt tuning and adapter-based methods.

---

> ### Author Response · Authors · 2024-11-15
> **Answer to reviewer NwVL**
>
> ### W1. In-depth analysis on effectiveness
> - By fine-tuning only the last visual projection matrix with supervision from text embeddings via cross-entropy loss, we argue that this strategy is closer in principle to CLIP's original pretraining regime. This is different from other methods that insert additional parameters and/or add a linear layer or MLP with regular classification supervision.
> - We also argue that the regularization that we impose to keep the trainable weights close to the pretrained ones helps in preserving the rich knowledge encapsulated in CLIP weights and in preventing overfitting and forgetting in the few-shot learning regime.
> - Towards a better understanding of ProLIP, we study and evaluate quantitatively ProLIP under different tasks (few-shot classification, domain generalization, cross-dataset generalization, test-time adaptation) and hyper-parameter configurations (grid search, with or without validation set, etc.)
>
>
> ### W2. Experiments with larger backbones, like ViT-Large
> - For fair comparison with prior works, in this paper we evaluated on multiple CLIP encoders (ResNet50, ResNet101, ViT-B/16, ViT-B/32) also considered in previous works.
> - We agree that it would be interesting to extend to larger backbones and we will consider it for future versions of this work.
>
>
> ### W3. Ablation on different shot numbers from 1-shot to N-shots
> - We consider the few-shot evaluation protocol from previous works with {1,2,4,8,16}-shot settings.
> - For all experiments and ablations on few-shot learning we consider all shot numbers, average over 11 datasets with 10 different support sets for each (see Tab. 1-2, Tab. 6-9, Fig. 2-3)
> - If the reviewer refers to the 4-shot protocol for domain generalization and cross-dataset generalization, we follow the protocol proposed in ProGrad for fair comparison.
> - Did the reviewer had other few-shot setting in mind?
>
>
> ### W4. Limitation on tasks with longer texts
> - We believe that in general CLIP is limited to short text sequences, as addressed by Zhang et al. (2024), and that all methods fine-tuning CLIP are likely to face such limitations whether using real text tokens or learnable ones.
>
>
> ### Q1. Could ProLIP be combined with prompt tuning method?
> - Thank you for the nice idea. Yes they can be combined. We will add this experiment in future work.
>
>
> ### Q2. How does ProLIP perform on datasets not represented in CLIP’s pretraining (e.g., medical or aerial imagery)?
> - Thank you for the question. Indeed we are interested in such datasets where fine-tuning would be necessary to steer CLIP features towards the downstream task.
> - Among the 11 datasets we study for few-shot classification there are the specific datasets EuroSAT (aerial imagery) and FGVCAircraft (fine-grained aircraft models) that are likely less represented in CLIP's pretraining and that we highlight in the text.
> - In the main paper we report scores aggregated over the 11 datasets, but we provide detailed results in Table 6-7. For example, ProLIP is top-performer on EuroSAT.
> - Furthermore, in cross-dataset generalization for EuroSAT (Table 4), ProLIP is again top-performer.
>
>
> ### Q3. Comparison of training and computation time
> - Thanks for the suggestion. We will include this in future work.
>
>
> **References**
> B. Zhang et al., Long-CLIP: Unlocking the Long-Text Capability of CLIP, ECCV 2024

---

### Author Response · Authors · 2024-11-15
**General response**

We thank the reviewers for providing insightful feedback, which will help to improve the paper.

After carefully reviewing the comments, we have decided to withdraw the paper.
This results also from a number of inaccurate or erroneous comments we have noticed in some reviews. In particular comments about: lack of diverse out-of-domain datasets, lack of ablation on shot numbers, sensitivity to the choice of regularization value $\lambda$, non state-of-the-art performance or state-of-the-art performance only on specific settings, lack of novelty, misalignment of results, fairness of hyper-parameter tuning.

We would like to respond to the comments and concerns expressed by the reviewers.

Please find our responses below in reply to individual reviews.

---

### Note · Authors · 2024-11-15

I have read and agree with the venue's withdrawal policy on behalf of myself and my co-authors.